# POLICY SMOOTHING FOR PROVABLY ROBUST REINFORCEMENT LEARNING

**Aounon Kumar, Alexander Levine, Soheil Feizi**
Department of Computer Science
University of Maryland - College Park, USA
`aounon@umd.edu,{alevine0,sfeizi}@cs.umd.edu`

## ABSTRACT

The study of provable adversarial robustness for deep neural networks (DNNs) has mainly focused on *static* supervised learning tasks such as image classification. However, DNNs have been used extensively in real-world *adaptive* tasks such as reinforcement learning (RL), making such systems vulnerable to adversarial attacks as well. Prior works in provable robustness in RL seek to certify the behaviour of the victim policy at every time-step against a non-adaptive adversary using methods developed for the static setting. But in the real world, an RL adversary can infer the defense strategy used by the victim agent by observing the states, actions, etc. from previous time-steps and adapt itself to produce stronger attacks in future steps (e.g., by focusing more on states critical to the agent's performance). We present an efficient procedure, designed specifically to defend against an adaptive RL adversary, that can directly certify the total reward without requiring the policy to be robust at each time-step. Focusing on randomized smoothing based defenses, our main theoretical contribution is to prove an *adaptive version* of the Neyman-Pearson Lemma – a key lemma for smoothing-based certificates – where the adversarial perturbation at a particular time can be a stochastic function of current and previous observations and states as well as previous actions. Building on this result, we propose *policy smoothing* where the agent adds a Gaussian noise to its observation at each time-step before passing it through the policy function. Our robustness certificates guarantee that the final total reward obtained by policy smoothing remains above a certain threshold, even though the actions at intermediate time-steps may change under the attack. We show that our certificates are *tight* by constructing a worst-case scenario that achieves the bounds derived in our analysis. Our experiments on various environments like Cartpole, Pong, Freeway and Mountain Car show that our method can yield meaningful robustness guarantees in practice.

## 1  INTRODUCTION

Deep neural networks (DNNs) have been widely employed for reinforcement learning (RL) problems as they enable the learning of policies directly from raw sensory inputs, like images, with minimal intervention from humans. From achieving super-human level performance in video-games (Mnih et al., 2013; Schulman et al., 2015; Mnih et al., 2016), Chess (Silver et al., 2017) and Go (Silver et al., 2016) to carrying out complex real-world tasks, such as controlling a robot (Levine et al., 2016) and driving a vehicle (Bojarski et al., 2016), deep-learning based algorithms have not only established the state of the art, but also become more effortless to train. However, DNNs have been shown to be susceptible to tiny malicious perturbations of the input designed to completely alter their predictions (Szegedy et al., 2014; Madry et al., 2018; Goodfellow et al., 2015). In the RL setting, an attacker may either directly corrupt the observations of an RL agent (Huang et al., 2017; Behzadan & Munir, 2017; Pattanaik et al., 2018) or act adversarially in the environment (Gleave et al., 2020) to significantly degrade the performance of the victim agent. Most of the adversarial defense literature has focused mainly on classification tasks (Kurakin et al., 2017; Buckman et al., 2018; Guo et al., 2018; Dhillon et al., 2018; Li & Li, 2017; Grosse et al., 2017; Gong et al., 2017).

In this paper, we study a defense procedure for RL problems that is provably robust against norm-bounded adversarial perturbations of the observations of the victim agent.

**Problem setup.** A reinforcement learning task is commonly described as a game between an agent and an environment characterized by the Markov Decision Process (MDP) $M = (S, A, T, R, \gamma)$, where $S$ is a set of states, $A$ is a set of actions, $T$ is the transition probability function, $R$ is the one-step reward function and $\gamma \in [0, 1]$ is the discount factor. However, as described in Section 3, our analysis applies to an even more general setting than MDPs. At each time-step $t$, the agent makes an observation $o_t = o(s_t) \in \mathbb{R}^d$ which is a probabilistic function of the current state of the environment, picks an action $a_t \in A$ and receives an immediate reward $R_t = R(s_t, a_t)$. We define an adversary as an entity that can corrupt the agent's observations of the environment by augmenting them with a perturbation $\epsilon_t$ at each time-step $t$ which can depend on the states, actions, observations, etc., generated so far. We use $\epsilon = (\epsilon_1, \epsilon_2, \ldots)$ to denote the entire sequence of adversarial perturbations. The goal of the adversary is to minimize the total reward obtained by the agent policy $\pi$ while keeping the overall $\ell_2$-norm of the perturbation within a budget $B$. Formally, the adversary seeks to optimize the following objective:

$$\min_{\epsilon} \mathbb{E}_{\pi} \left[ \sum_{t=0}^{\infty} \gamma^t R_t \right], \text{ where } R_t = R(s_t, a_t), a_t \sim \pi(\cdot | o(s_t) + \epsilon_t)$$

$$\text{s.t. } \|(\epsilon_1, \epsilon_2, \ldots)\|_2 = \sqrt{\sum_{t=0}^{\infty} \|\epsilon_t\|_2^2} \leq B.$$

Note that the size of the perturbation $\epsilon_t$ in each time-step $t$ need not be the same and the adversary may choose to distribute the budget $B$ over different time-steps in a way that allows it to produce a stronger attack. Also, our formulation accounts for cases when the agent may only partially observe the state of the environment, making $M$ a Partially Observable Markov Decision Process (POMDP).

**Objective.** Our goal in provably robust RL is to design a policy $\pi$ such that the total reward in the presence of a norm-bounded adversary is guaranteed to remain above a certain threshold, i.e.,

$$\min_{\epsilon} \mathbb{E}_{\pi} \left[ \sum_{t=0}^{\infty} \gamma^t R_t \right] \geq \underline{R}, \text{ s.t. } \|\epsilon\|_2 \leq B. \tag{1}$$

In other words, no norm-bounded adversary can lower the expected total reward of the policy $\pi$ below a certain threshold. In our discussion, we restrict out focus to finite-step games that end after $t$ time-steps. This is a reasonable approximation for infinite games with $\gamma < 1$, as for a sufficiently large $t$, $\gamma^t$ becomes negligibly small. For games where $\gamma = 1$, $R_t$ must become sufficiently small after a finite number of steps to keep the total reward finite.

**Step-wise vs. episodic certificates**. Previous works on robust RL have sought to certify the behaviour of the policy function at *each* time-step of an episode, e.g., the output of a Deep Q-Network (Lütjens et al., 2019) and the action taken for a given state (Zhang et al., 2020). Ensuring that the behaviour of the policy remains unchanged in each step can also certify that the final total reward remains the same under attack. However, if the per-step guarantee fails at even one of the intermediate steps, the certificate on the total reward becomes vacuous or impractical to compute (as noted in Appendix E of Zhang et al. (2020)). Our approach gets around this issue by directly certifying the final total reward for the entire episode *without* requiring the policy to be provably robust at each intermediate step. Also, the threat-model we consider is more general as we allow the adversary to choose the size of the perturbation for each time-step. Thus, our method can defend against more sophisticated attacks that focus more on states that are crucial for the victim agent's performance.

**Technical contributions.** In this paper, we study a defense procedure based on "randomized smoothing" (Cohen et al., 2019; Lécuyer et al., 2019; Li et al., 2019; Salman et al., 2019) since at least in "static" settings, its robustness guarantee scales up to high-dimensional problems and does not need to make stringent assumptions about the model. We ask: *can we utilize the benefits of randomized smoothing to make a general high-dimensional RL policy provably robust against adversarial attacks?* The answer to this question turns out to be non-trivial as the adaptive nature of the adversary in the RL setting makes it difficult to apply certificates from the static setting. For example, the $\ell_2$-certificate by Cohen et al. (2019) critically relies on the clean and adversarial distributions

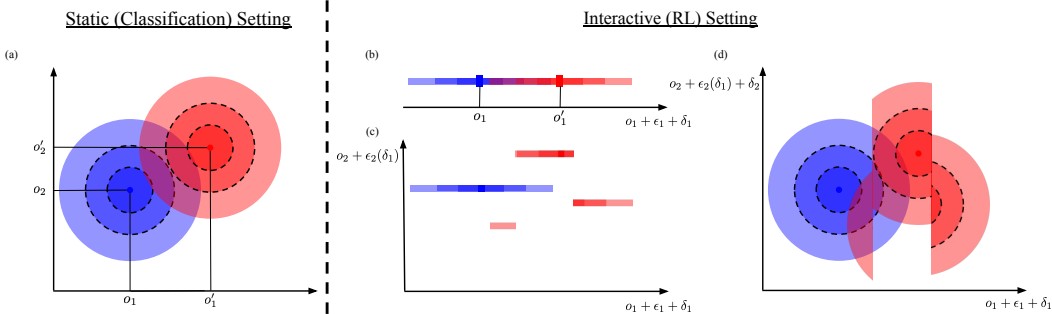

Figure 1: The standard Cohen et al. (2019) smoothing-based robustness certificate relies on the clean and the adversarial distributions being isometric Gaussians (panel a). However, adding noise to sequential observations in an RL setting (panels b-d) *does not* result in an isometric Gaussian distribution over the space of observations. In all figures, the distributions associated with clean and adversarially-perturbed values are shown in blue and red, respectively.

being isometric Gaussians (Figure 1-a). However, in the RL setting, the adversarial perturbation in one step might depend on states, actions, observations, etc., of the previous steps, which could in turn depend on the random Gaussian noise samples added to the observations in these steps. Thus, the resulting adversarial distribution need not be isometric as in the static setting (Figure 1-(b-d)). For more details on this example, see Appendix B.

Our main theoretical contribution is to prove an *adaptive version* of the Neyman-Pearson lemma (Neyman & Pearson, 1992) to produce robustness guarantees for RL. We emphasize that this is *not* a straightforward extension (refer to Appendix D, E and F for the entire proof). To prove this fundamental result, we first eliminate the effect of randomization in the adversary (Lemma 1) by converting a general adversary to one where the perturbation at each time-step is a deterministic function of the previous states, actions, observations, etc., and showing that the modified adversary is as strong as the general one. Then, we prove the adaptive Neyman-Pearson lemma where we show that, in the worst-case, the deterministic adversary can be converted to one that uses up the entire budget $B$ in the first coordinate of the perturbation in the first time-step (Lemma 3). Finally, we derive the robustness guarantee under an isometric Gaussian smoothing distribution (Theorem 1). In section A, we establish the *tightness* of our certificates by constructing the worst-case environment-policy pair which attains our derived bounds. More formally, out of all the environment-policy pairs that achieve a certain total reward with probability $p$, we show a worst-case environment-policy pair and a corresponding adversary such that the probability of achieving the same reward under the presence of the adversary is minimum. A discussion on the Neyman-Pearson lemma in the context of randomized smoothing is available in Appendix C.

Building on these theoretical results, we propose **Policy Smoothing**, a simple model-agnostic randomized-smoothing based technique that can provide certified robustness without increasing the computational complexity of the agent's policy. Our main contribution is to show that by augmenting the policy's input by a random smoothing noise, we can achieve provable robustness guarantees on the total reward under a norm-bounded adversarial attack (Section 4.2). Policy Smoothing does not need to make assumptions about the agent's policy function and is also oblivious to the workings of RL environment. Thus, this method can be applied to any RL setting without having to make restrictive assumptions on the environment or the agent. In section 3, we model the entire adversarial RL process under Policy Smoothing as a sequence of interactions between a system **A**, which encapsulates the RL environment and the agent, and a system **B**, which captures the addition of the adversarial perturbation and the smoothing noise to the observations. Our theoretical results do not require these systems to be Markovian and can thus have potential applications in real-time decision-making processes that do not necessarily satisfy the Markov property.

**Empirical Results.** We use four standard Reinforcement Learning benchmark tasks to evaluate the effectiveness of our defense and the significance of our theoretical results: the Atari games 'Pong' and 'Freeway' (Mnih et al., 2013) and the classical 'Cartpole' and 'Mountain Car' control environments (Barto et al., 1983; Moore, 1990) – see Figure 3. We find that our method provides highly nontrivial certificates. In particular, on at least two of the tasks, 'Pong' and 'cartpole', the

*provable lower bounds* on the average performances of the defended agents, against any adversary, exceed the observed average performances of undefended agents under a practical attack.

## 2  PRIOR WORK

**Adversarial RL.** Adversarial attacks on RL systems have been extensively studied in recent years. DNN-based policies have been attacked by either directly corrupting their inputs (Huang et al., 2017; Behzadan & Munir, 2017; Pattanaik et al., 2018) or by making adversarial changes in the environment (Gleave et al., 2020). Empirical defenses based on adversarial training, whereby the dynamics of the RL system is augmented with adversarial noise, have produced good results in practice (Kamalaruban et al., 2020; Vinitsky et al., 2020). Zhang et al. (2021) propose training policies together with a learned adversary in an online alternating fashion to achieve robustness to perturbations of the agent's observations.

**Robust RL.** Prior work by Lütjens et al. (2019) has proposed a 'certified' defense against adversarial attacks to observations in deep reinforcement learning, particularly for Deep Q-Network agents. However, that work essentially only guarantees the stability of the *network approximated Q-value* at each time-step of an episode. By contrast, our method provides a bound on the expected *true reward* of the agent under any norm-bounded adversarial attack.

Zhang et al. (2020) certify that the action in each time-step remains unchanged under an adversarial perturbation of fixed budget for every time-step. This can guarantee that the final total reward obtained by the robust policy remains the same under attack. However, this approach would not be able to yield any robustness certificate if even one of the intermediate actions changed under attack. Our approach gets around this difficulty by directly certifying the total reward, letting some of the intermediate actions of the robust policy to potentially change under attack. For instance, consider an RL agent playing Atari Pong. The actions taken by the agent when the ball is close to and approaching the paddle are significantly more important than the ones when the ball is far away or retreating from the paddle. By allowing some of the intermediate actions to potentially change, our approach can certify for larger adversarial budgets and provide a more fine-grained control over the desired total-reward threshold. Moreover, we study a more general threat model where the adversary may allocate different attack budgets for each time-step focusing more on the steps that are crucial for the agent's performance, e.g., attacking a Pong agent when the ball is close to the paddle.

**Provable Robustness in Static Settings:** Notable provable robustness methods in static settings are based on interval-bound propagation (Gowal et al., 2018; Huang et al., 2019; Dvijotham et al., 2018; Mirman et al., 2018), curvature bounds (Wong & Kolter, 2018; Raghunathan et al., 2018; Chiang et al., 2020; Singla & Feizi, 2019; 2020; 2021), randomized smoothing (Cohen et al., 2019; Lécuyer et al., 2019; Li et al., 2019; Salman et al., 2019; Levine & Feizi, 2021), etc. Certified robustness has also been extended to problems with structured outputs such as images and sets (Kumar & Goldstein, 2021). Focusing on Gaussian smoothing, Cohen et al. (2019) showed that if a classifier outputs a class with some probability under an isometric Gaussian noise around an input point, then it will output that class with high probability at any perturbation of the input within a particular $\ell_2$ distance. Kumar et al. (2020) showed how to certify the expectation of softmax scores of a neural network under Gaussian smoothing by using distributional information about the scores.

## 3  PRELIMINARIES AND NOTATIONS

We model the finite-step adversarial RL framework as a $t$-round communication between two systems **A** and **B** (Figure 2). System **A** represents the RL game. It contains the environment $M$ and the agent, and when run independently, simulates the interactions between the two for some given policy $\pi$. At each time-step $i$, it generates a *token* $\tau_i$ from some set $\mathcal{T}$, which is a tuple of the current state $s_i$ and its observation $o_i$, the action $a_{i-1}$ in the previous step (and potentially some other objects that we ignore in this discussion), i.e., $\tau_i = (s_i, a_{i-1}, o_i, \ldots) \in S \times A \times \mathbb{R}^d \times \ldots = \mathcal{T}$. For the first step, replace the action in $\tau_1$ with some dummy element $*$ from the action space $A$. System **B** comprises of the adversary and the smoothing distribution which generate an adversarial perturbation $\epsilon_i$ and a smoothing noise vector $\delta_i$, respectively, at each time-step $i$, the sum of which is denoted by an *offset* $\eta_i = \epsilon_i + \delta_i \in \mathbb{R}^d$.

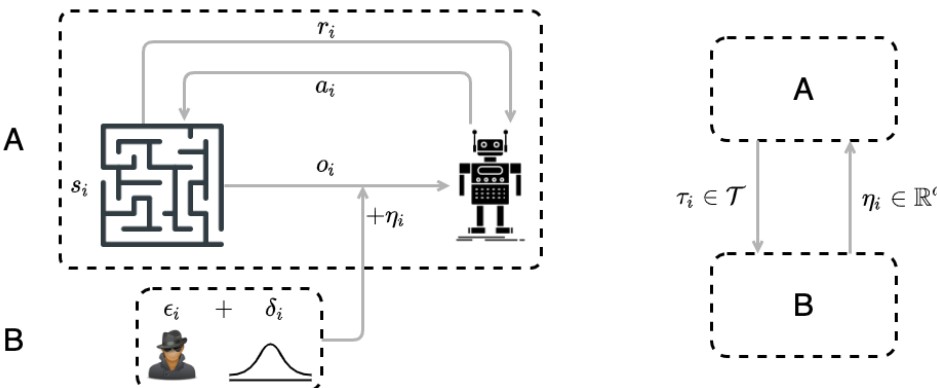

Figure 2: Adversarial robustness framework.

When both systems are run together in an interactive fashion, in each round $i$, system **A** generates $\tau_i$ as a probabilistic function of $\tau_1, \eta_1, \tau_2, \eta_2, \ldots, \tau_{i-1}, \eta_{i-1}$, i.e., $\tau_i : (\mathcal{T} \times \mathbb{R}^d)^{i-1} \to \Delta(\mathcal{T})$. $\tau_1$ is sampled from a fixed distribution. It passes $\tau_i$ to **B**, which generates $\epsilon_i$ as a probabilistic function of $\{\tau_j, \eta_j\}_{j=1}^{i-1}$ and $\tau_i$, i.e., $\epsilon_i : (\mathcal{T} \times \mathbb{R}^d)^{i-1} \times \mathcal{T} \to \Delta(\mathbb{R}^d)$ and adds a noise vector $\delta_i$ sampled independently from the smoothing distribution to obtain $\eta_i$. It then passes $\eta_i$ to **A** for the next round. After running for $t$ steps, a deterministic or random 0/1-function $h$ is computed over all the tokens and offsets generated. We are interested in bounding the probability with which $h$ outputs 1 as a function of the adversarial budget $B$. In the RL setting, $h$ could be a function indicating whether the total reward is above a certain threshold or not.

## 4 PROVABLY ROBUST RL

### 4.1 ADAPTIVE NEYMAN-PEARSON LEMMA

Let $X$ be the random variable representing the tuple $z = (\tau_1, \eta_1, \tau_2, \eta_2, \ldots, \tau_t, \eta_t) \in (\mathcal{T} \times \mathbb{R}^d)^t$ when there is no adversary, i.e., $\epsilon_i = 0$ and $\eta_i = \delta_i$ is sampled directly form the smoothing distribution $\mathcal{P}$. Let $Y$ be the random variable representing the same tuple in the presence of a general adversary $\epsilon$ satisfying $\|\epsilon\|_2 \leq B$. Thus, if $h(X) = 1$ with some probability $p$, we are interested in deriving a lower-bound on the probability of $h(Y) = 1$ as a function of $p$ and $B$. Let us now define a deterministic adversary $\epsilon^{dt}$ for which the adversarial perturbation at each step is a deterministic function of the tokens and offsets of the previous steps and the token generated in the current step. i.e., $\epsilon_i^{dt} : (\mathcal{T} \times \mathbb{R}^d)^{i-1} \times \mathcal{T} \to \mathbb{R}^d$. Let $Y^{dt}$ be its corresponding random variable. Then, we have the following lemma that converts a probabilistic adversary into a deterministic one.

**Lemma 1** (**Reduction to Deterministic Adversaries**). *For any general adversary $\epsilon$ and an $\Gamma \subseteq (\mathcal{T} \times \mathbb{R}^d)^t$, there exists a deterministic adversary $\epsilon^{dt}$ such that,*

$$\mathbb{P}[Y^{dt} \in \Gamma] \leq \mathbb{P}[Y \in \Gamma],$$

*where $Y^{dt}$ is the random variable for the distribution defined by the adversary $\epsilon^{dt}$.*

This lemma says that for any adversary (deterministic or random) and a subset $\Gamma$ of the space of $z$, there exists a deterministic adversary which assigns a lower probability to $\Gamma$ than the general adversary. In the RL setting, this means that the probability with which a smoothed policy achieves a certain reward value under a general adversary is lower-bounded by the probability of the same under a deterministic adversary. The intuition behind this lemma is that out of all the possible values that the internal randomness of the adversary may assume, there exists a sequence of values that assigns the minimum probability to $\Gamma$ (over the randomness of the environment, policy, smoothing noise, etc.). We defer the proof to the appendix.

Next, we formulate an adaptive version of the Neyman-Pearson lemma for the case when the smoothing distribution $\mathcal{P}$ is an isometric Gaussian $\mathcal{N}(0, \sigma^2 I)$. If we applied the classical Neyman-Pearson lemma on the distributions of $X$ and $Y^{dt}$, it will give us a characterization of the worst-case 0/1 function among the class of functions that achieve a certain probability $p$ of being 1 under the distribution of $X$ that has the minimum probability of being 1 under $Y^{dt}$. Let $\mu_X$ and $\mu_{Y^{dt}}$ be the probability density function of $X$ and $Y^{dt}$, respectively.

**Lemma 2** (**Neyman-Pearson Lemma, 1933**). *If* $\Gamma_{Y^{dt}} = \{z \in (\mathcal{T} \times \mathbb{R}^d)^t \mid \mu_{Y^{dt}}(z) \leq q\mu_X(z)\}$ *for some* $q \geq 0$ *and* $\mathbb{P}[h(X) = 1] \geq \mathbb{P}[X \in \Gamma_{Y^{dt}}]$, *then* $\mathbb{P}[h(Y^{dt}) = 1] \geq \mathbb{P}[Y^{dt} \in \Gamma_{Y^{dt}}]$.

For an arbitrary element $h$ in the class of functions $H_p = \{h \mid \mathbb{P}[h(X) = 1] \geq p\}$, construct the set $\Gamma_{Y^{dt}}$ for an appropriate value of $q$ for which $\mathbb{P}[X \in \Gamma_{Y^{dt}}] = p$. Now, consider a function $h'$ which is 1 if its input comes from $\Gamma_{Y^{dt}}$ and 0 otherwise. Then, the above lemma says that the function $h'$ has the minimum probability of being 1 under $Y^{dt}$, i.e.,

$$h' = \operatorname*{argmin}_{h \in H_p} \mathbb{P}[h(Y^{dt}) = 1].$$

This gives us the worst-case function that achieves the minimum probability under an adversarial distribution. However, in the adaptive setting, $\Gamma_{Y^{dt}}$ could be a very complicated set and obtaining an expression for $\mathbb{P}[Y^{dt} \in \Gamma_{Y^{dt}}]$ might be difficult. To simplify our analysis, we construct a *structured* deterministic adversary $\epsilon^{st}$ which exhausts its entire budget in the first coordinate of the first perturbation vector, i.e., $\epsilon_1^{st} = (B, 0, \ldots, 0)$ and $\epsilon_i^{st} = (0, 0, \ldots, 0)$ for $i > 1$. Let $Y^{st}$ be the corresponding random variable and $\mu_{Y^{st}}$ its density function. We formulate the following adaptive version of the Neyman-Pearson lemma:

**Lemma 3** (**Adaptive Neyman-Pearson Lemma**). *If* $\Gamma_{Y^{st}} = \{z \in (\mathcal{T} \times \mathbb{R}^d)^t \mid \mu_{Y^{st}}(z) \leq q\mu_X(z)\}$ *for some* $q \geq 0$ *and* $\mathbb{P}[h(X) = 1] \geq \mathbb{P}[X \in \Gamma_{Y^{st}}]$, *then* $\mathbb{P}[h(Y^{dt}) = 1] \geq \mathbb{P}[Y^{st} \in \Gamma_{Y^{st}}]$.

The key difference from the classical version is that the worst-case set we construct in this lemma is for the structured adversary and the final inequality relates the probability of $h$ outputting 1 under the adaptive adversary to the probability that the structured adversary assigns to the worst-case set. It says that for the appropriate value of $q$ for which $\mathbb{P}[X \in \Gamma_{Y^{st}}] = p$, any function $h \in H_p$ outputs 1 with at least the probability that $Y^{st}$ assigns to $\Gamma_{Y^{st}}$. It shows that over all possible functions in $H_p$ and over all possible adversaries $\epsilon$, the indicator function $\mathbf{1}_{z \in \Gamma_{Y^{st}}}$ and the structured adversary capture the worst-case scenario where probability of $h$ being 1 under the adversarial distribution is the minimum. Since both $Y^{st}$ and $X$ are just isometric Gaussian distribution with the same variance $\sigma^2$ centered at different points on the first coordinate of $\eta_1$, the set $\Gamma_{Y^{st}}$ is the set of all tuples $z$ for which $\{\eta_1\}_1$ is below a certain threshold.[1] We use lemmas 1 and 3 to derive the final bound on the probability of $h(Y) = 1$ in the following theorem, the proof of which is deferred to the appendix.

**Theorem 1** (**Robustness Guarantee**). *For an isometric Gaussian smoothing noise with variance* $\sigma^2$, *if* $\mathbb{P}[h(X) = 1] \geq p$, *then:*

$$\mathbb{P}[h(Y) = 1] \geq \Phi(\Phi^{-1}(p) - B/\sigma),$$

*where* $\Phi$ *is the standard normal CDF.*

The above analysis can be adapted to obtain an upper-bound on $\mathbb{P}[h(Y) = 1]$ of $\Phi(\Phi^{-1}(p) + B/\sigma)$.

## 4.2 Policy Smoothing

Building on these results, we develop *policy smoothing*, a simple model-agnostic randomized-smoothing based technique that can provide certified robustness without increasing the computational complexity of the agent's policy. Given a policy $\pi$, we define a smoothed policy $\bar{\pi}$ as:

$$\bar{\pi}\left( \cdot \mid o(s_t) \right) = \pi\left( \cdot \mid o(s_t) + \delta_t \right), \text{ where } \delta_t \sim \mathcal{N}(0, \sigma^2 I).$$

Our goal is to certify the expected sum of the rewards collected over multiple time-steps under policy $\bar{\pi}$. We modify the technique developed by Kumar et al. (2020) to certify the expected class scores of a neural network by using the empirical cumulative distribution function (CDF) of the scores under the smoothing distribution to work for the RL setting. This approach utilizes the fact that the expected value of a random variable $\mathcal{X}$ representing a class score under a Gaussian $\mathcal{N}(0, \sigma^2 I)$ smoothing noise can be expressed using its CDF $F(.)$ as below:

$$\mathbb{E}[\mathcal{X}] = \int_0^\infty (1 - F(x))dx - \int_{-\infty}^0 F(x)dx. \tag{2}$$

---

[1] We use $\{\eta_i\}_j$ to denote the $j$th coordinate of the vector $\eta_i$.

(a) Cartpole          (b) Pong          (c) Freeway          (d) Mountain Car

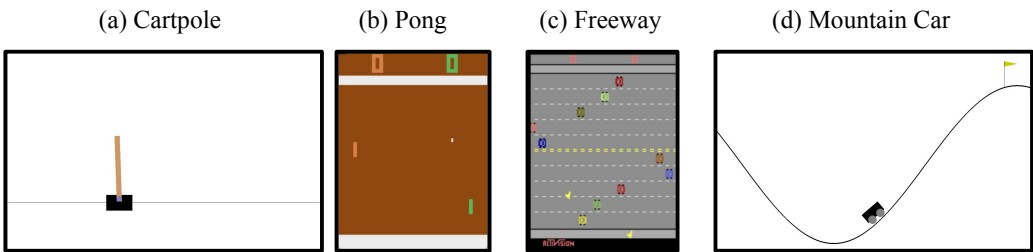

Figure 3: Environments used in evaluations rendered by OpenAI Gym (Brockman et al., 2016).

Given $m$ samples $\{x_i\}_{i=1}^m$ of the random variable $\mathcal{X}$, let us define its empirical CDF at a point $x$, $F_m(x) = |\{x_i \mid x_i \leq x\}|/m$, as the fraction of samples that are less than or equal to $x$. Using $F_m(x)$, the Dvoretzky–Kiefer–Wolfowitz inequality can produce high-confidence bounds on the true CDF of $\mathcal{X}$. It says that with probability $1 - \alpha$, for $\alpha \in (0, 1]$, the true CDF $F(x)$ is in the range $[\underline{F}(x), \overline{F}(x)]$, where $\underline{F}(x) = F_m(x) - \sqrt{\ln(2/\alpha)/2m}$ and $\overline{F}(x) = F_m(x) + \sqrt{\ln(2/\alpha)/2m}$. For an adversarial perturbation of $\ell_2$-size $B$, the result of Cohen et al. (2019) bounds the CDF within $[\Phi(\Phi^{-1}(\underline{F}(x)) - B/\sigma), \Phi(\Phi^{-1}(\overline{F}(x)) + B/\sigma)]$, which in turn bounds $E[\mathcal{X}]$ using equation (2).

In the RL setting, we can model the total reward as a random variable and obtain its empirical CDF by playing the game using policy $\bar{\pi}$. As above, we can bound the CDF $F(x)$ of the total reward in a range $[\underline{F}(x), \overline{F}(x)]$ using the empirical CDF. Applying Theorem 1, we can bound the CDF within $[\Phi(\Phi^{-1}(\underline{F}(x)) - B/\sigma), \Phi(\Phi^{-1}(\overline{F}(x)) + B/\sigma)]$ for an $\ell_2$ adversary of size $B$. The function $h$ in Theorem 1 could represent the CDF $F(x)$ by indicating whether the total reward computed for an input $z \in (\mathcal{T} \times \mathbb{R}^d)^t$ is below a value $x$. Finally, equation (2) puts bounds on the expected total reward under an adversarial attack.

## 5   EXPERIMENTS

### 5.1   ENVIRONMENTS AND SETUP

We tested on four standard environments: the classical cortrol problems 'Cartpole' and 'Mountain Car' and the Atari games 'Pong' and 'Freeway.' We consider three tasks which use a discrete action space ('Cartpole' and the two Atari games) as well as one task that uses a continuous action space ('Mountain Car'). For the discrete action space tasks, we use a standard Deep Q-Network (DQN) (Mnih et al., 2013) model, while for 'Mountain Car', we use Deep Deterministic Policy Gradient (DDPG) (Lillicrap et al., 2016).

As is common in DQN and DDPG, our agents choose actions based on multiple frames of observations. In order to apply a realistic threat model, we assume that the adversary acts on each frame *only once* when it is first observed. The adversarial distortion is then maintained when the same frame is used in future time-steps. In other words, we consider the observation at time step $o_t$ (discussed in Section 3) to be only the *new* observation at time $t$: this means that the adversarial/noise perturbation $\eta_t$, as a *fixed* vector, continues to be used to select the next action for several subsequent time-steps. This is a realistic model because we are assuming that the adversary can affect the agent's observation of states, not necessarily the agent's *memory* of previous observations. As in other works on smoothing-based defenses (e.g., Cohen et al. (2019)), we add noise during training as well as at test time. We use DQN and DDPG implementations from the popular stable-baselines3 package (Raffin et al., 2019): hyperparameters are provided in the appendix. In experiments, we report and certify for the total *non-discounted* ($\gamma = 1$) reward.

In 'Cartpole', the observation vector consists of four kinematic features. We use a simple MLP model for the Q-network, and tested two variations: one in which the agent uses five frames of observation, and one in which the agent uses only a single frame (shown in the appendix).

In order to show the effectiveness of our technique on tasks involving high-dimensional state observations, we chose two tasks ('Pong' and 'Freeway') from the Atari environment, where state observations are image frames, observed as $84 \times 84$ pixel greyscale images. For 'Pong', we test on a "one-round" variant of the original environment. In our variant, the game ends after one player, either the agent or the opponent, scores a goal: the reward is then either zero or one. Note that this is *not* a one-timestep episode: it takes typically on the order of 100 timesteps for this to occur. Results

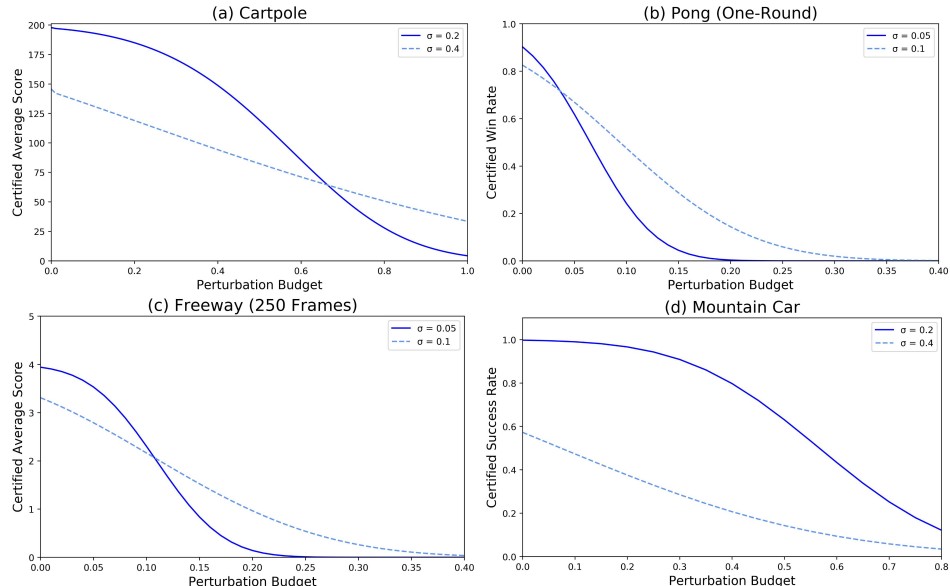

Figure 4: Certified performance for various environments. The certified lower-bound on the mean reward is based on a 95% lower confidence interval estimate of the mean reward of the smoothed model, using 10,000 episodes.

for a full Pong game are presented in the appendix: as explained there, we find that the certificates unfortunately do not scale with the length of the game. For the 'Freeway' game, we play on 'Hard' mode and end the game after 250 timesteps.

In order to test on an environment with a *continuous* action space, we chose the 'Mountain Car' environment. Note that previous certification results for reinforcement learning, which certify actions at individual states rather than certifying the overall reward (Zhang et al., 2020) *cannot* be applied to continuous action state problems. In this environment, the observation vector consists of two kinematic features (position and velocity), and the action is one continuous scalar (acceleration). As in 'Cartpole', we use five observation frames and a simple MLP policy. We use a slight variant of the original environment: we do not penalize for fuel cost so the reward is a boolean representing whether or not the car reaches the destination in the time allotted (999 steps).

## 5.2 RESULTS

Certified lower bounds on the expected total reward, as a function of the total perturbation budget, are presented in Figure 4. For tasks with zero-one total reward ('Pong' and 'Mountain Car'), the function to be smoothed represents the total reward: $h(\cdot) = R$ where $R$ is equal to 1 if the agent wins the round, and 0 otherwise. To compute certificates on games with continuous scores ('Cartpole' and 'Freeway'), we use CDF smoothing (Kumar et al., 2020): see appendix for technical details.

In order to evaluate the robustness of both undefended and policy-smoothed agents, we developed an attack tailored to the threat model defined in 1, where the adversary makes a perturbation to state observations which is *bounded over the entire episode*. For DQN agents, as in Lütjens et al. (2019), we perturb the observation $o$ such that the perturbation-induced action $a' := \arg\max_a Q(o + \epsilon_t, a)$ minimizes the (network-approximated) Q-value of the true observation $Q(o, a')$. However, in order to conserve adversarial budget, we *only* attack if the gap between attacked q-value $Q(o, a')$ and the clean q-value $\max_a Q(o, a)$ is sufficiently large, exceeding a preset threshold $\lambda_Q$. In practice, this allows the attacker to concentrate the attack budget only on the time-steps which are critical to the agent's performance. When attacking DDPG, where both a Q-value network and a policy network $\pi$ are trained and the action is taken according to $\pi$, we instead minimize $Q(o, \pi(o + \epsilon_t)) + \lambda \|\epsilon_t\|^2$ where the hyperparameter $\lambda$ plays an analogous role in focusing perturbation budget on "important" steps, as judged by the effect on the approximated Q-value. Empirical results are presented in Figure 5. We see that the attacks are effective on the undefended agents (red, dashed lines). In fact, from comparing Figures 4 and 5, we see that, for the Pong and Cartpole environments, the undefended performance under attack is worse than the *certified lower bound* on the performance of the policy-smoothed agents under *any possible* attack: our certificates are the clearly non-vacuous for these environments. Further details on the attack optimizations are provided in the appendix.

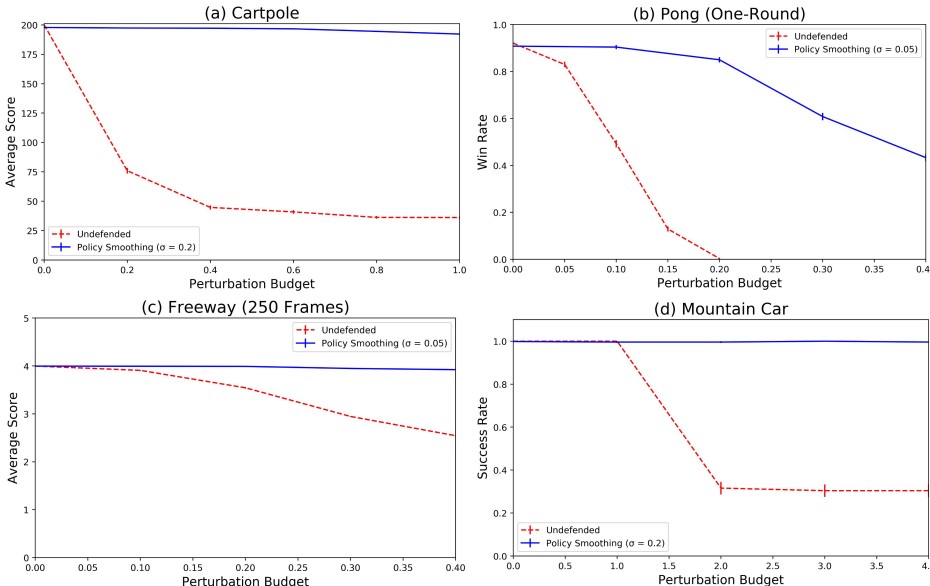

Figure 5: Empirical robustness of defended and undefended agents. Full details of attacks are presented in appendix.

We also present an attempted empirical attack on the smoothed agent, adapting techniques for attacking smoothed classifiers from Salman et al. (2019) (solid blue lines). We observed that our model was highly robust to this attack – significantly more robust than guaranteed by our certificate. However, it is not clear whether this is due to looseness in the certificate or to weakness of the attack: the significant practical challenges to attacking smoothed agents are also discussed in the appendix.

## 6 CONCLUSION

In this work, we extend randomized smoothing to design a procedure that can make any reinforcement learning agent provably robust against adversarial attacks without significantly increasing the complexity of the agent's policy. We show how to adapt existing theory on randomized smoothing from static tasks such as classification, to the dynamic setting of RL. By proving an adaptive version of the celebrated Neyman-Pearson Lemma, we show that by adding Gaussian smoothing noise to the input of the policy, one can certifiably defend it against norm-bounded adversarial perturbations of its input. The policy smoothing technique and its theory covers a wide range of adversaries, policies and environments. Our analysis is tight, meaning that the certificates we achieve are best possible unless restrictive assumptions about the RL game are made. In our experiments, we show that our method provides meaningful guarantees on the robustness of the defended policies and the total reward they achieve even in the worst case is higher than an undefended policy. In the future, the introduction of randomized smoothing to RL could inspire the design of provable robustness techniques for control problems in dynamic real-world environments and multi-agent RL settings.

## REPRODUCIBILITY

We supplement our work with accompanying code for reproducing the experimental results, as well as pre-trained models for a selection of the experiments. Details about setting hyper-parameters and the environments we test are included in the appendix. Proofs for our theoretical results (lemmas and main theorem) are also provided in the appendix.

## ETHICS STATEMENT

We present a method to make RL models provably robust under adversarial perturbations. We do not foresee any immediate ethical concerns associated with our work.

## ACKNOWLEDGEMENTS

This project was supported in part by NSF CAREER AWARD 1942230, a grant from NIST 60NANB20D134, HR001119S0026-GARD-FP-052, HR00112090132, ONR YIP award N00014-22-1-2271, Army Grant W911NF2120076.

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

## A    Tightness of the Certificate

Here, we present a worst-case environment-policy pair that achieves the bound in Theorem 1, showing that our robustness certificate is in fact tight. For a given environment $M = (S, A, T, R, \gamma)$ and a policy $\pi$, let $p$ be a lower-bound on the probability that the total reward obtained by policy $\pi$ under Gaussian smoothing (no adversary) with variance $\sigma^2$ is above a certain threshold $\nu$, i.e.,

$$\mathbb{P}\left[\sum_{i=1}^{t} \gamma^{i-1} R_i \geq \nu\right] \geq p.$$

Let $H_p$ be the class of all such environment-policy pairs that cross this reward threshold with probability at least $p$. We construct an environment-policy pair $(M', \pi')$ that achieves the reward threshold $\nu$ with probability $\Phi(\Phi^{-1}(p) - B/\sigma)$ under the structured adversary $\epsilon^{st}$. Note that, this does not mean that $\epsilon^{st}$ is the strongest possible adversary for a general environment-policy pair. It only shows that the performance of policy $\pi'$ in environment $M'$ under the adversary $\epsilon^{st}$ is a lower-bound on the performance of a general environment-policy pair under a general adversary. Consider a one-step game with environment $M' = (S, A, T', R', \gamma)$ with a deterministic observation function $o$ of the state-space and a policy $\pi'$ such that $\pi'$ returns an action $a_1 \in A$ if the first coordinate of $o(s_1) + \eta_1$ is at most $\omega = \{o(s_1)\}_1 + \sigma \Phi^{-1}(p)$ and another action $a_2 \in A$ otherwise. Here $\{o(s_1)\}_1$ represents the first coordinate of $o(s_1)$. The environment offers a reward $\nu$ if the action in the first step is $a_1$ and 0 when it is $a_2$. The game terminates immediately. The probability of the reward being above $\nu$ is equal to the probability of the action being $a_1$. When $\eta_1$ is sampled from the Gaussian distribution, this probability is equal to $\Phi((\omega - \{o(s_1)\}_1)/\sigma) = p$. Therefore, $(M', \pi') \in H_p$. Under the presence of the structured adversary $\epsilon^{st}$ defined in Section 4.1, this probability after smoothing becomes $\Phi((\omega - \{o(s_1)\}_1 - B)/\sigma) = \Phi(\Phi^{-1}(p) - B/\sigma)$, which is same as the bound in Theorem 1.

## B    Static Vs. Adaptive Setting

In this section, we illustrate the difference between the adversarial distributions in the static setting and the adaptive setting. Naively, one might assume that smoothing-based robustness guarantees can be applied directly to reinforcement learning, by adding noise to observations. For example, it seems plausible to use Cohen et al.'s $\ell_2$ certificate Cohen et al. (2019), which relies on the overlap in the distributions of isometric Gaussians with different means, by simply adding Gaussian noise to each observation (Figure 1-a). However, as we demonstrate with a toy example in Figure 1-(b-d), the Cohen et al. certificate *cannot* be applied directly to the RL setting, because adding noise to sequential observations *does not* result in an isometric Gaussian distribution over the space of observations. This is because the adversarial offset to later observations may be conditioned on the noise added to previous observations. In 1-(b-d), we consider a two-step episode, and for simplicity, we consider a case where the ground-truth observations at each step are fixed. At step 1, the noised distributions of the clean observation $o_1$ and the adversarially-perturbed observation $o_1'$ are both Gaussians and overlap substantially, similar to in the standard classification setting (panel b). However, we see in panel (c) that the adversarial perturbation $\epsilon_2$ added to $o_2$ *can depend on* the smoothed value of $o_1'$. This is because the agent may leak information about the observation that it receives after smoothing ($o_1 + \eta_1$) to the adversary, for example through its choice of actions. After smoothing is performed on $o_2$, the adaptive nature of the adversary causes the distribution of smoothed observations to no longer be an isometric Gaussian in the adversarial case (panel d). The standard certification results therefore cannot be applied.

## C    Neyman–Pearson lemma [1993] in Smoothing

In the context of randomized smoothing, the Neyman–Pearson lemma produces the worst-case decision boundary of a classifier based on the estimated probability of the top class under the smoothing distribution. It says that this boundary is a region where the ratio of the probability density functions of the smoothing distributions at the clean input and the perturbed input is a constant. When the two distributions are isometric Gaussians, as is the case in static settings like image classification, this boundary takes the form of a hyper-plane (see Appendix A of Cohen et al. (2019)). However, in the dynamic setting of RL, the smoothing distribution after adding the adversarial perturbation may not

be isometric even if the smoothing noise at each time-step was sampled from an isometric Gaussian distribution (see figure 1, section 'Technical contributions' and Appendix A). So, we formulate and prove an adaptive version of the Neyman-Pearson lemma to obtain provable robustness in RL through randomized smoothing.

## D  PROOF OF LEMMA 1

**Statement:** *For any general adversary $\epsilon$ and an $\Gamma \subseteq (\mathcal{T} \times \mathbb{R}^d)^t$, there exists a deterministic adversary $\epsilon^{dt}$ such that,*

$$\mathbb{P}[Y^{dt} \in \Gamma] \leq \mathbb{P}[Y \in \Gamma],$$

*where $Y^{dt}$ is the random variable for the distribution defined by the adversary $\epsilon^{dt}$.*

*Proof.* Consider a time-step $j$ such that $\forall i < j, \epsilon_i$ is a deterministic function of $\tau_1, \eta_1, \tau_2, \eta_2, \ldots, \tau_{i-1}, \eta_{i-1}, \tau_i$. Let $\mathcal{H} = \{z \mid z_1 = \tau_1, z_2 = \eta_1, z_3 = \tau_2, z_4 = \eta_2, \ldots, z_{2j-1} = a_j\}$ be the set of points whose first $2j-1$ coordinates are fixed to an arbitrary set of values $\tau_1, \eta_1, \tau_2, \eta_2, \ldots, \tau_j$. In the space defined by $\mathcal{H}$, $\epsilon_1, \ldots, \epsilon_{j-1}$ are fixed vectors in $\mathbb{R}^d$ and $\epsilon_j$ is sampled from a fixed distribution over the vectors with $\ell_2$-norm at most $B_r^j$. Let $Y_{\mathcal{H}}^\gamma$ be the random variable representing the distribution over points in $\mathcal{H}$ defined by the adversary for which $\epsilon_j = \gamma$, such that $\|\gamma\|_2 \leq B_r^j$. Define an adversary $\epsilon'$, such that, $\epsilon'_i = \epsilon_i, \forall i \neq j$. Set $\epsilon'_j$ to the vector $\gamma$ that minimizes the probability that $Y_{\mathcal{H}}^\gamma$ assigns to $\Gamma \cap \mathcal{H}$, i.e.,

$$\epsilon'_j = \underset{\|\gamma\|_2 \leq B_r^j}{\arg\min} \, \mathbb{P}[Y_{\mathcal{H}}^\gamma \in \Gamma \cap \mathcal{H}]$$

The adversary $\epsilon'$ behaves as $\epsilon$ up to step $j-1$. At step $j$, it sets $\epsilon'_j$ to the $\gamma$ that minimizes the probability it assigns to $\Gamma \cap \mathcal{H}$, based on the values $\tau_1, \eta_1, \tau_2, \eta_2, \ldots, \tau_j$. After that, it mimics $\epsilon$ till the last time-step $t$. Therefore, for a given tuple $(z_1, z_2, \ldots, z_{2j-1}) = (\tau_1, \eta_1, \tau_2, \eta_2, \ldots, \tau_j)$,

$$\mathbb{P}[Y_{\mathcal{H}}^{\epsilon'_j} \in \Gamma \cap \mathcal{H}] \leq \mathbb{P}[Y \in \Gamma \cap \mathcal{H}]$$

Since both adversaries are same up to step $j-1$, their respective distributions over $z_1, z_2, \ldots, z_{2j-1}$ remains same as well. Therefore, integrating both sides of the above inequality over the space of all tuples $(z_1, z_2, \ldots, z_{2j-1})$, we have:

$$\int \mathbb{P}[Y_{\mathcal{H}}^{\epsilon'_j} \in \Gamma \cap \mathcal{H}] p_Y(z_1, z_2, \ldots, z_{2j-1}) dz_1 dz_2 \ldots dz_{2j-1}$$

$$\leq \int \mathbb{P}[Y \in \Gamma \cap \mathcal{H}] p_Y(z_1, z_2, \ldots, z_{2j-1}) dz_1 dz_2 \ldots dz_{2j-1}$$

$$\implies \mathbb{P}[Y' \in \Gamma] \leq \mathbb{P}[Y \in \Gamma],$$

where $Y'$ is the random variable corresponding to $\epsilon'$. Thus, we have constructed an adversary where the first $j$ adversarial perturbations are a deterministic function of the $\tau_i$s and $\eta_i$s of the previous rounds. Applying the above step sufficiently many times we can construct a deterministic adversary $\epsilon^{dt}$ represented by the random variable $Y^{dt}$ such that

$$\mathbb{P}[Y^{dt} \in \Gamma] \leq \mathbb{P}[Y \in \Gamma].$$

$\square$

## E  PROOF OF LEMMA 3

Lemma 3 states that the structured adversary characterises the worst-case scenario. Before proving this lemma, let us first show that any deterministic adversary can be converted to one that uses up the entire budget of $B$ without increasing the probability it assigns to $h$ being one in the worst-case. For each step $i$, let us define a *used budget* $B_u^i = \|(\epsilon_1, \epsilon_2, \ldots, \epsilon_{i-1})\|_2$ as the norm of the perturbations of the previous steps and a *remaining budget* $B_r^i = \sqrt{B^2 - (B_u^i)^2}$ as an upper-bound on the norm of the perturbations of the remaining steps. Note that, $B_u^1 = 0$ and $B_r^1 = B$.

Consider a version $\tilde{\epsilon}^{dt}$ of the deterministic adversary that uses up the entire available budget $B$ by scaling up $\epsilon_t^{dt}$ such that its norm is equal to $B_r^t$, i.e., setting it to $\epsilon_t^{dt} B_r^t / \|\epsilon_t^{dt}\|_2$. Let $\tilde{Y}^{dt}$ be the random variable representing $\tilde{\epsilon}^{dt}$.

**Lemma.** *If* $\Gamma_{\tilde{Y}^{dt}} = \{z \in (\mathcal{T} \times \mathbb{R}^d)^t \mid \mu_{\tilde{Y}^{dt}}(z) \leq q\mu_X(z)\}$ *for some* $q \geq 0$ *and* $\mathbb{P}[h(X) = 1] \geq \mathbb{P}[X \in \Gamma_{\tilde{Y}^{dt}}]$, *then* $\mathbb{P}[h(Y^{dt}) = 1] \geq \mathbb{P}[\tilde{Y}^{dt} \in \Gamma_{\tilde{Y}^{dt}}]$.

*Proof.* Consider $\Gamma_{Y^{dt}} = \{z \in (\mathcal{T} \times \mathbb{R}^d)^t \mid \mu_{Y^{dt}}(z) \leq q'\mu_X(z)\}$ for some $q' \geq 0$, such that, $\mathbb{P}[X \in \Gamma_{Y^{dt}}] = p$ for some lower-bound $p$ on $\mathbb{P}[h(X) = 1]$. Then, by the Neyman-Pearson Lemma we have that,

$$\mathbb{P}[h(Y^{dt}) = 1] \geq \mathbb{P}[Y^{dt} \in \Gamma_{Y^{dt}}].$$

Now consider a space $\mathcal{H}$ in $(\mathcal{T} \times \mathbb{R}^d)^t$ where all but the last element of the tuple $z$ are fixed, i.e., $\mathcal{H} = \{z \mid z_1 = \tau_1, z_2 = \eta_1, z_3 = \tau_2, z_4 = \eta_2, \ldots, z_{2t-1} = \tau_t\}$ Since, $\epsilon^{dt}$ is a deterministic adversary where each $\epsilon_i^{dt}$ is a deterministic function of the previous $\tau_i$s and $\eta_i$s, each $\epsilon_i^{dt}$ is also fixed in $\mathcal{H}$. Therefore, in $\mathcal{H}$, both $\mu_X$ and $\mu_{Y^{dt}}$ are two isometric Gaussians in the space of the $\eta_i$s and the set $\mathcal{H} \cap \Gamma_{Y^{dt}}$ is a hyperplane. The probability assigned by $Y^{dt}$ to $\mathcal{H} \cap \Gamma_{Y^{dt}}$ is proportional to the distance of the center of the corresponding Gaussian. In the construction of $\tilde{\epsilon}^{dt}$, this distance can only increase, therefore,

$$\mathbb{P}[Y^{dt} \in \Gamma_{Y^{dt}}] \geq \mathbb{P}[\tilde{Y}^{dt} \in \Gamma_{Y^{dt}}]$$

Now, consider a function $h_{\Gamma_{Y^{dt}}}(z)$ which outputs one if $z \in \Gamma_{Y^{dt}}$ and zero otherwise. Construct the set $\Gamma_{\tilde{Y}^{dt}} = \{z \in (\mathcal{T} \times \mathbb{R}^d)^t \mid \mu_{\tilde{Y}^{dt}}(z) \leq q\mu_X(z)\}$ for some $q \geq 0$ such that,

$$\mathbb{P}[X \in \Gamma_{\tilde{Y}^{dt}}] = p = \mathbb{P}[h_{\Gamma_{Y^{dt}}}(X) = 1].$$

Then, by the Neyman-Pearson Lemma, we have,

$$\mathbb{P}[h_{\Gamma_{Y^{dt}}}(\tilde{Y}^{dt}) = 1] \geq \mathbb{P}[\tilde{Y}^{dt} \in \Gamma_{\tilde{Y}^{dt}}]$$

$$\text{or,} \quad \mathbb{P}[\tilde{Y}^{dt} \in \Gamma_{Y^{dt}}] \geq \mathbb{P}[\tilde{Y}^{dt} \in \Gamma_{\tilde{Y}^{dt}}] \qquad \text{(from definition of } h_{\Gamma_{Y^{dt}}})$$

$$\text{or,} \quad \mathbb{P}[Y^{dt} \in \Gamma_{Y^{dt}}] \geq \mathbb{P}[\tilde{Y}^{dt} \in \Gamma_{\tilde{Y}^{dt}}]$$

$$\text{or,} \quad \mathbb{P}[h(Y^{dt}) = 1] \geq \mathbb{P}[\tilde{Y}^{dt} \in \Gamma_{\tilde{Y}^{dt}}], \qquad \text{(from the above two inequalities)}$$

proving the statement of the lemma. $\qquad\square$

Now, we prove lemma 3 below:

**Statement:** *If* $\Gamma_{Y^{st}} = \{z \in (\mathcal{T} \times \mathbb{R}^d)^t \mid \mu_{Y^{st}}(z) \leq q\mu_X(z)\}$ *for some* $q \geq 0$ *and* $\mathbb{P}[h(X) = 1] \geq \mathbb{P}[X \in \Gamma_{Y^{st}}]$, *then* $\mathbb{P}[h(Y^{dt}) = 1] \geq \mathbb{P}[Y^{st} \in \Gamma_{Y^{st}}]$.

*Proof.* Construct the set $\Gamma_{\tilde{Y}^{dt}}$ as defined in the above lemma for a $q \geq 0$ such that $\mathbb{P}[X \in \Gamma_{\tilde{Y}^{dt}}] = p$, for some lower-bound $p$ on $\mathbb{P}[h(X) = 1]$. Then,

$$\mathbb{P}[h(Y^{dt}) = 1] \geq \mathbb{P}[\tilde{Y}^{dt} \in \Gamma_{\tilde{Y}^{dt}}]$$

Now consider the structured adversary $\epsilon^{st}$ in which $\epsilon_1^{st} = (B, 0, \ldots, 0)$ and $\epsilon_i^{st} = (0, 0, \ldots, 0)$ for $i > 1$. Define the set $\Gamma_{Y^{st}} = \{z \in (\mathcal{T} \times \mathbb{R}^d)^t \mid \mu_{Y^{st}}(z) \leq q\mu_X(z)\}$ for the same $q$ as above. Then, we can show that:

1. $\mathbb{P}[\tilde{Y}^{dt} \in \Gamma_{\tilde{Y}^{dt}}] = \mathbb{P}[Y^{st} \in \Gamma_{Y^{st}}]$, and

2. $\mathbb{P}[X \in \Gamma_{\tilde{Y}^{dt}}] = \mathbb{P}[X \in \Gamma_{Y^{st}}]$

which, in turn, prove the statement of the lemma.

Let $\mathcal{N}$ and $\mathcal{N}_{\epsilon_i}$ represent Gaussian distributions centered at origin and $\epsilon_i$ respectively. Then, we can write $\mu_X$ and $\mu_Y$ as below:

$$\mu_X(z) = \prod_{i=1}^t \mu_{T_i}(\tau_i \mid \tau_1, \eta_1, \tau_2, \eta_2, \ldots, \tau_{i-1}, \eta_{i-1})\mu_{\mathcal{N}}(\eta_i)$$

$$\mu_{\tilde{Y}^{dt}}(z) = \prod_{i=1}^t \mu_{T_i}(\tau_i \mid \tau_1, \eta_1, \tau_2, \eta_2, \ldots, \tau_{i-1}, \eta_{i-1})\mu_{\mathcal{N}_{\tilde{\epsilon}_i^{dt}}}(\eta_i)$$

where $\mu_{T_i}$ is the conditional probability distribution of token $\tau_i$ given the previous tokens and offsets. Therefore,

$$\frac{\mu_{\tilde{Y}^{dt}}(z)}{\mu_X(z)} = \prod_{i=1}^{t} \frac{\mu_{\mathcal{N}_{\tilde{\epsilon}_i^{dt}}}(\eta_i)}{\mu_{\mathcal{N}}(\eta_i)} = \prod_{i=1}^{t} e^{\frac{\eta_i^T \eta_i - (\eta_i - \tilde{\epsilon}_i^{dt})^T(\eta_i - \tilde{\epsilon}_i^{dt})}{2\sigma^2}}$$

$$\frac{\mu_{\tilde{Y}^{dt}}(z)}{\mu_X(z)} \leq q \iff \sum_{i=1}^{t} 2\eta_i^T \tilde{\epsilon}_i^{dt} - (\tilde{\epsilon}_i^{dt})^T \tilde{\epsilon}_i^{dt} \leq 2\sigma^2 \ln q$$

Consider a round $j \leq t$ such that $\tilde{\epsilon}_i^{dt} = 0, \forall i > j+1$ and $\tilde{\epsilon}_{j+1}^{dt} = (B_r^{j+1}, 0, \ldots, 0)$. We can always find such a $j$ as we always have $\tilde{\epsilon}_{t+1}^{dt} = (B_r^{t+1}, 0, \ldots, 0)$, since $B_r^{t+1} = 0$. Note that, $B_r^{j+1} = \sqrt{B^2 - \left(B_u^{j+1}\right)^2}$ and in turn $\tilde{\epsilon}_{j+1}^{dt}$ are functions of $\tau_1, \eta_1, \tau_2, \eta_2, \ldots, \tau_j$ and not $\tau_{j+1}$. Let $\mathcal{H} = \{z \mid z_1 = \tau_1, z_2 = \eta_1, z_3 = \tau_2, z_4 = \eta_2, \ldots, z_{2j-1} = \tau_j\}$ be the set of points whose first $2j - 1$ coordinates are fixed to an arbitrary set of values $\tau_1, \eta_1, \tau_2, \eta_2, \ldots, \tau_j$. For points in $\mathcal{H}$, all $\tilde{\epsilon}_i^{dt}$ for $i \leq j+1$ are fixed and for $i > j+1$ are set to zero. Let $\tilde{Y}_{\mathcal{H}}^{dt}$ denote the random variable representing the distribution of points in $\mathcal{H}$ defined by the adversary $\tilde{\epsilon}^{dt}$ (corresponding random variable $\tilde{Y}^{dt}$). In the space of $\eta_j, \eta_{j+1}, \ldots, \eta_t$, this is an isometric Gaussian centered at $(\tilde{\epsilon}_j^{dt}, \tilde{\epsilon}_{j+1}^{dt}, 0, \ldots, 0)$. Therefore, $\Gamma \cap \mathcal{H}$ is given by

$$\sum_{i=1}^{j+1} 2\eta_i^T \tilde{\epsilon}_i^{dt} - (\tilde{\epsilon}_i^{dt})^T \tilde{\epsilon}_i^{dt} \leq 2\sigma^2 \ln t$$

$$\text{or,} \quad \eta_j^T \tilde{\epsilon}_j^{dt} + \eta_{j+1}^T \tilde{\epsilon}_{j+1}^{dt} \leq \beta, \tag{3}$$

for some constant $\beta$ dependent on $\eta_1, \tilde{\epsilon}_1^{dt}, \ldots, \eta_{j-1}, \tilde{\epsilon}_{j-1}^{dt}, \sigma$ and $t$. The probability assigned by the Gaussian random variable $Y_{\mathcal{H}}$ to the half-space defined by (3) is proportional to the distance of the center of the Gaussian from the hyper-plane in (3), which is equal to:

$$\frac{\|\tilde{\epsilon}_j^{dt}\|^2 + \|\tilde{\epsilon}_{j+1}^{dt}\|^2 - \beta}{\sqrt{\|\tilde{\epsilon}_j^{dt}\|^2 + \|\tilde{\epsilon}_{j+1}^{dt}\|^2}} = \frac{(B_r^j)^2 - \beta}{B_r^j},$$

where the equality follows from:

$$\begin{aligned}
\|\tilde{\epsilon}_j^{dt}\|^2 + \|\tilde{\epsilon}_{j+1}^{dt}\|^2 &= \|\tilde{\epsilon}_j^{dt}\|^2 + (B_r^{j+1})^2 \\
&= \|\tilde{\epsilon}_j^{dt}\|^2 + B^2 - (B_u^{j+1})^2 & \text{(from definition of } B_r^i) \\
&= \|\tilde{\epsilon}_j^{dt}\|^2 + B^2 - (\|\tilde{\epsilon}_1^{dt}\|^2 + \|\tilde{\epsilon}_2^{dt}\|^2 + \ldots + \|\tilde{\epsilon}_j^{dt}\|^2) \\
&= B^2 - (\|\tilde{\epsilon}_1^{dt}\|^2 + \|\tilde{\epsilon}_2^{dt}\|^2 + \ldots + \|\tilde{\epsilon}_{j-1}^{dt}\|^2) \\
&= B^2 - (B_u^j)^2 = (B_r^j)^2.
\end{aligned}$$

Now, consider an adversary $\tilde{\epsilon}^{dt'}$ such that $\tilde{\epsilon}_i^{dt'} = \tilde{\epsilon}_i^{dt}, \forall i \leq j-1$, $\tilde{\epsilon}_j^{dt'} = (B_r^j, 0, \ldots, 0)$, and $\tilde{\epsilon}_i^{dt} = 0, \forall i > j$. Let $\tilde{Y}^{dt'}$ be the corresponding random variable. Define $\Gamma_{\tilde{Y}^{dt'}}$ similar to $\Gamma_{\tilde{Y}^{dt}}$. Then, $\Gamma_{\tilde{Y}^{dt'}} \cap \mathcal{H}$ is given by

$$\eta_j^T(B_r^j, 0, \ldots, 0) \leq \beta, \tag{4}$$

which is obtained by replacing $\tilde{\epsilon}_j^{dt}$ with $(B_r^j, 0, \ldots, 0)$ and $\tilde{\epsilon}_{j+1}^{dt}$ with $(0, 0, \ldots, 0)$ in inequality (3) about the origin. Define $\tilde{Y}_{\mathcal{H}}^{dt'}$ similar to $\tilde{Y}_{\mathcal{H}}^{dt}$, and just like $\tilde{Y}_{\mathcal{H}}^{dt}$, the distribution of $\tilde{Y}_{\mathcal{H}}^{dt'}$ is also an isometric Gaussian, but is centered at $((B_r^j, 0, \ldots, 0), (0, 0, \ldots, 0))$. The probability assigned by this Gaussian distribution to $\Gamma_{\tilde{Y}^{dt'}} \cap \mathcal{H}$ is proportional to the distance of its center to the hyper-plane defining the region in (4), which is equal to $((B_r^j)^2 - \beta)/B_r^j$. Therefore,

$$\mathbb{P}[\tilde{Y}_{\mathcal{H}}^{dt} \in \Gamma_{\tilde{Y}^{dt}} \cap \mathcal{H}] = \mathbb{P}[\tilde{Y}_{\mathcal{H}}^{dt'} \in \Gamma_{\tilde{Y}^{dt'}} \cap \mathcal{H}].$$

The key intuition behind this step is that, for isometric Gaussian smoothing distribution, the worst-case probability assigned by the adversarial distribution only depends on the magnitude of the perturbation and not its direction. Figure 6 illustrates this property for a two-dimensional input space.

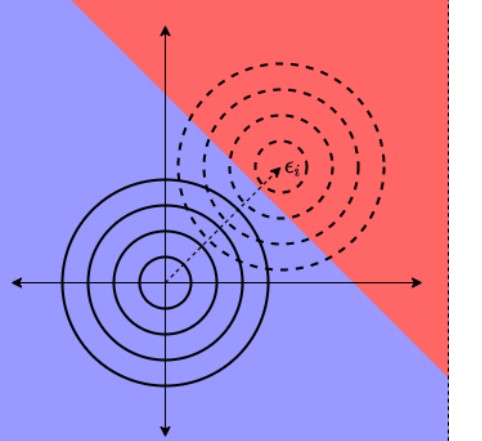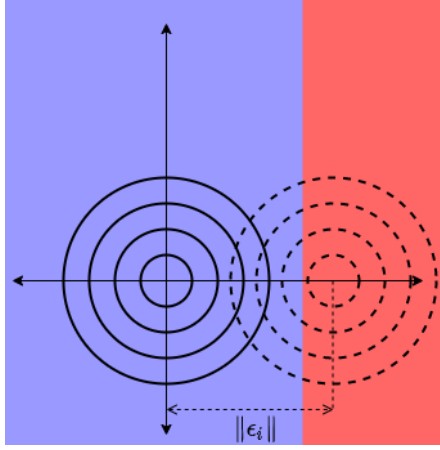

Figure 6: General adversarial perturbation vs. perturbation aligned along the first dimension. Blue and red regions denote where the worst-case function is one and zero respectively.

Since both adversaries are same up to step $j-1$, their respective distributions over $z_1, z_2, \ldots, z_{2j-1}$ remains same as well, i.e., $p_{\tilde{Y}^{dt}}(z_1, z_2, \ldots, z_{2j-1}) = p_{\tilde{Y}^{dt'}}(z_1, z_2, \ldots, z_{2j-1})$. Integrating over the space of all tuples $(z_1, z_2, \ldots, z_{2j-1})$, we have:

$$\int \mathbb{P}[\tilde{Y}^{dt}_{\mathcal{H}} \in \Gamma_{\tilde{Y}^{dt}} \cap \mathcal{H}] p_{\tilde{Y}^{dt}}(z_1, z_2, \ldots, z_{2j-1}) dz_1 dz_2 \ldots dz_{2j-1}$$

$$= \int \mathbb{P}[\tilde{Y}^{dt'}_{\mathcal{H}} \in \Gamma_{\tilde{Y}^{dt'}} \cap \mathcal{H}] p_{\tilde{Y}^{dt'}}(z_1, z_2, \ldots, z_{2j-1}) dz_1 dz_2 \ldots dz_{2j-1}$$

$$\implies \mathbb{P}[\tilde{Y}^{dt}_{\mathcal{H}} \in \Gamma_{\tilde{Y}^{dt}}] = \mathbb{P}[\tilde{Y}^{dt'}_{\mathcal{H}} \in \Gamma_{\tilde{Y}^{dt'}}],$$

Since the distribution defined by $X$ (with no adversary) over the space of $\eta_i$s is a Gaussian centered at origin whose distance to both $\Gamma_{\tilde{Y}^{dt}} \cap \mathcal{H}$ and $\Gamma_{\tilde{Y}^{dt'}} \cap \mathcal{H}$ is the same (equal to $-\beta/B_r^j$), it assigns the same probability to both (3) and (4). Therefore,

$$\mathbb{P}[X \in \Gamma_{\tilde{Y}^{dt}}] = \mathbb{P}[X \in \Gamma_{\tilde{Y}^{dt'}}].$$

Thus, we have constructed an adversary with one less non-zero $\epsilon_i$. Applying, this step sufficiently many times we can obtain the adversary $\epsilon^{st}$ such that,

$$\mathbb{P}[\tilde{Y}^{dt} \in \Gamma_{\tilde{Y}^{dt}}] = \mathbb{P}[Y^{st} \in \Gamma_{Y^{st}}] \quad \text{and} \quad \mathbb{P}[X \in \Gamma_{\tilde{Y}^{dt}}] = \mathbb{P}[X \in \Gamma_{Y^{st}}]$$

which completes the proof. $\qquad \square$

## F  PROOF OF THEOREM 1

**Statement:** *For an isometric Gaussian smoothing noise with variance $\sigma^2$, if $\mathbb{P}[h(X) = 1] \geq p$, then:*

$$\mathbb{P}[h(Y) = 1] \geq \Phi(\Phi^{-1}(p) - B/\sigma).$$

*Proof.* Define $\Gamma_Y = \{z \in (\mathcal{T} \times \mathbb{R}^d)^t \mid \mu_Y(z) \leq q\mu_X(z)\}$ for an appropriate $q$ such that $\mathbb{P}[X \in \Gamma_Y] = p$. Then, by the Neyman-Pearson lemma, we have $\mathbb{P}[h(Y) = 1] \geq \mathbb{P}[Y \in \Gamma_Y]$. Applying lemma 1, we know that there exists a deterministic adversary $\epsilon^{dt}$ represented by random variable $Y^{dt}$, such that,

$$\mathbb{P}[h(Y) = 1] \geq \mathbb{P}[Y \in \Gamma_Y] \geq \mathbb{P}[Y^{dt} \in \Gamma_Y]. \tag{5}$$

Now define a function $h_{\Gamma_Y}(z) = \mathbf{1}_{\{z \in \Gamma_Y\}}$ and a set $\Gamma_{Y^{dt}} = \{z \in (\mathcal{T} \times \mathbb{R}^d)^t \mid \mu_{Y^{dt}}(z) \leq q'\mu_X(z)\}$ for an appropriate $q' > 0$, such that, $\mathbb{P}[X \in \Gamma_{Y^{dt}}] = \mathbb{P}[h_{\Gamma_Y}(X) = 1] = p$. Applying the Neyman-

Pearson lemma again, we have:

$$\mathbb{P}[h_{\Gamma_Y}(Y^{dt}) = 1] \geq \mathbb{P}[Y^{dt} \in \Gamma_{Y^{dt}}]$$

$$\text{or,} \quad \mathbb{P}[Y^{dt} \in \Gamma_Y] \geq \mathbb{P}[Y^{dt} \in \Gamma_{Y^{dt}}] \qquad \text{(from definition of } h_{\Gamma_Y})$$

$$\text{or,} \quad \mathbb{P}[h(Y) = 1] \geq \mathbb{P}[Y^{dt} \in \Gamma_{Y^{dt}}] \qquad \text{(from inequality (5))}$$

Define $h_{\Gamma_{Y^{dt}}}(z) = \mathbf{1}_{\{z \in \Gamma_{Y^{dt}}\}}$. For the structured adversary $\epsilon^{st}$ represented by $Y^{st}$, define $\Gamma_{Y^{st}} = \{z \in (\mathcal{T} \times \mathbb{R}^d)^t \mid \mu_{Y^{st}}(z) \leq q'' \mu_X(z)\}$ for an appropriate $q'' > 0$, such that, $\mathbb{P}[X \in \Gamma_{Y^{st}}] = \mathbb{P}[h_{\Gamma_{Y^{dt}}}(X) = 1] = p$. Applying lemma 3, we have:

$$\mathbb{P}[h_{\Gamma_{Y^{dt}}}(Y^{dt}) = 1] \geq \mathbb{P}[Y^{st} \in \Gamma_{Y^{st}}]$$

$$\mathbb{P}[Y^{dt} \in \Gamma_{Y^{dt}}] \geq \mathbb{P}[Y^{st} \in \Gamma_{Y^{st}}] \qquad \text{(from definition of } h_{\Gamma_{Y^{dt}}})$$

$$\mathbb{P}[h(Y) = 1] \geq \mathbb{P}[Y^{st} \in \Gamma_{Y^{st}}] \qquad \text{(since } \mathbb{P}[h(Y) = 1] \geq \mathbb{P}[Y^{dt} \in \Gamma_{Y^{dt}}])$$

$\Gamma_{Y^{st}}$ is defined as the set of points $z$ which satisfy:

$$\frac{\mu_{Y^{st}}(z)}{\mu_X(z)} \leq q'' \quad \text{or,} \quad \frac{\mu_{\mathcal{N}_{\bar{\epsilon}_1^{st}}}(\eta_1)}{\mu_{\mathcal{N}}(\eta_1)} \leq q''$$

$$\eta_1^T(B, 0, \ldots, 0) \leq \beta \quad \text{or,} \quad \{\eta_1\}_1 \leq \beta/B$$

for some constant $\beta$. This is the set of all tuples $z$ where the first coordinate of $\eta_1$ is below a certain threshold $\gamma$. Since $\mathbb{P}[X \in \Gamma_{Y^{st}}] = p$,

$$\Phi(\gamma/\sigma) = p \implies \gamma = \sigma\Phi^{-1}(p).$$

Therefore,

$$\mathbb{P}[Y^{st} \in \Gamma_{Y^{st}}] = \Phi\left(\frac{\gamma - B}{\sigma}\right) = \Phi(\Phi^{-1}(p) - B/\sigma).$$

$\square$

# G   ADDITIONAL CARTPOLE RESULTS

We performed two additional experiment on Cartpole: we tested at larger noise levels, ($\sigma = 0.6$ and $0.8$) and we tested a variant of the agent architecture. Specifically, in addition to the agent shown in the main text, which uses five frames of observation, we also tested an agent which uses only a single frame. Unlike the Atari environment, the task is in fact solvable (in the non-adversarial case) using only one frame: the observation vector represents the complete system state. We computed certificates for the policy-smoothed version of this model, and tested attacks on the undefended version. (We did not test attacks on the smoothed single-frame variant). As we see in Figure 7, we achieve non-vacuous certificates in both settings (i.e, at large perturbation sizes, the smoothed agent is guaranteed to be more robust than the empirical robustness of a non-smoothed agent). However, observe that the undefended agent in the multi-frame setting is much more vulnerable to adversarial attack. This is likely because the increased number of total features (20 vs. four) introduces more complexity of the Q-network, making it more vulnerable against adversarial attack.

# H   FULL PONG GAME

In Figure 8, we explore a failure case of our technique: we fail to produce non-vacuous certificates for a full Pong game, where the game ends after either player scores 21 goals. In particular, while, for the one-round Pong game, the smoothed agent is provably more robust than the empirical performance of the undefended agent, this is clearly not the case for the full game. To understand why our certificate is vacuous here, note that in the the "worst-case" environment that our certificate assumes, any perturbation will (maximally) affect all future rewards. However, in the multi-round Pong game, each round of the game is only loosely coupled to the previous rounds (the ball momentum – but not position – as well as the paddle positions are retained). Therefore, any perturbation can only have a very limited effect on the total reward. Another way to think about this is to recall that in smoothing-based certificates, the noise added to *each* feature is proportional to the *total* perturbation budget of the adversary. In this sort of serial game, the perturbation budget required to attack the average reward scales with the (square root of the) number of rounds, but the noise tolerance of the agent does not similarly scale.

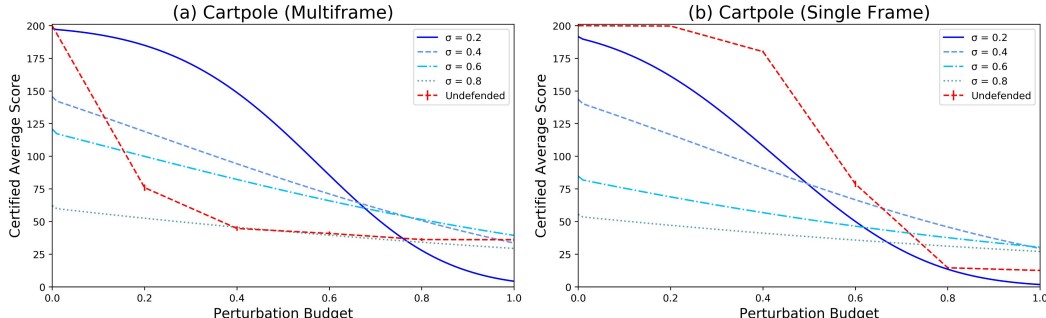

Figure 7: Additional Cartpole results. Attacks on smoothed agents at all $\sigma$ for the multiframe agents are presented in Appendix J

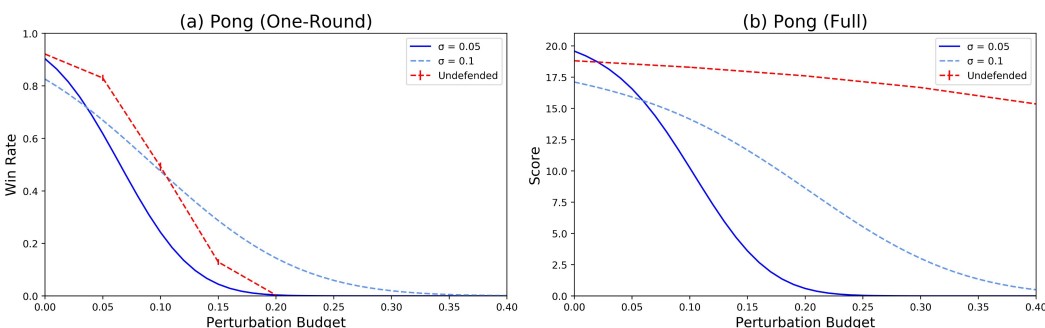

Figure 8: Results for the Full Pong game, compared to the single-round game.

## I TRAINING AND CLEAN TEST RESULTS

In Figure 9, we present the clean (non-attacked) test performance for the experiments presented in the main text, as a function of the smoothing noise $\sigma$.

In Figure 10, we present the clean training (i.e., validation round) performance as a function of the training time step and the smoothing noise $\sigma$. Note that early stopping was applied: the model from the best validation round was kept, and only replaced if a strictly better validation performance was recorded later.

- For Cartpole: logs were not kept after the first time an evaluation round had a perfect average score of 200 (this is because the "best model" was saved for this evaluation, and it would be impossible to beat this score, so training was not continued). However, for other tasks (i.e. mountain car) logs continued after a perfect evaluation round.

- For Freeway: as mentioned in Appendix Section L, we trained 5 times at each noise level, and kept the best of all 5 models. All 5 training curves are shown here for each noise level.

## J COMPLETE ATTACK RESULTS

In Figures 11 and 12, we report the empirical robustness under attack for all tested values of $\lambda_Q$: in the main text, we show only the result for the $\lambda_Q$ that represents the strongest attack. Figure 12 also shows the attacks on smoothed agents for all smoothing noises. All attack results are means over 1000 episodes (except for Mountain Car results, where 250 episodes were used) and error bars represent the standard error of the mean.

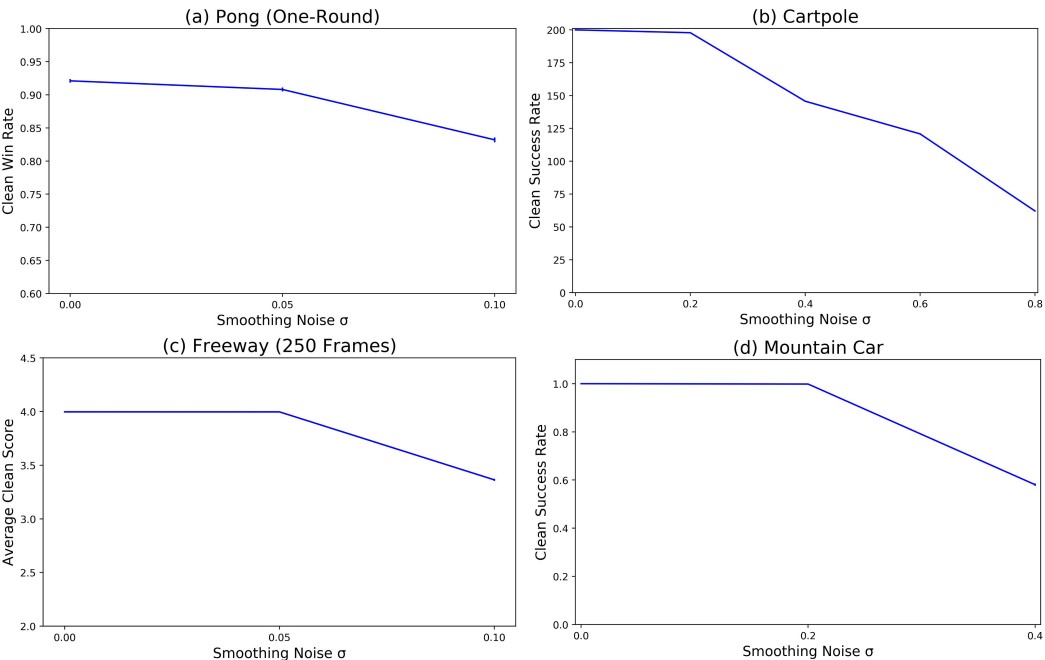

Figure 9: Clean test performance as a function of smoothing noise $\sigma$.

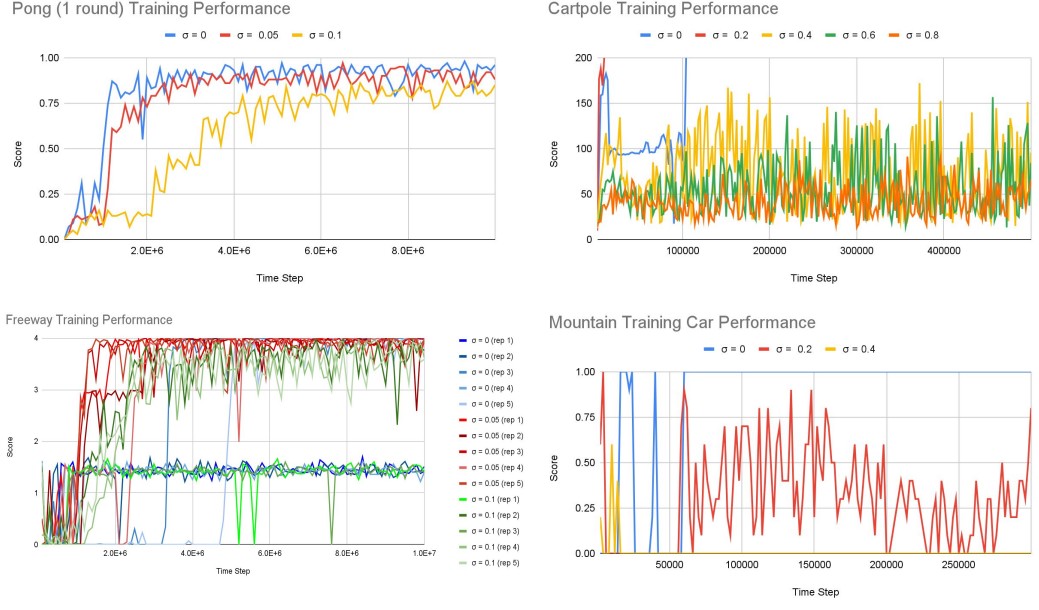

Figure 10: Clean training performance as a function of smoothing noise $\sigma$ and training step.

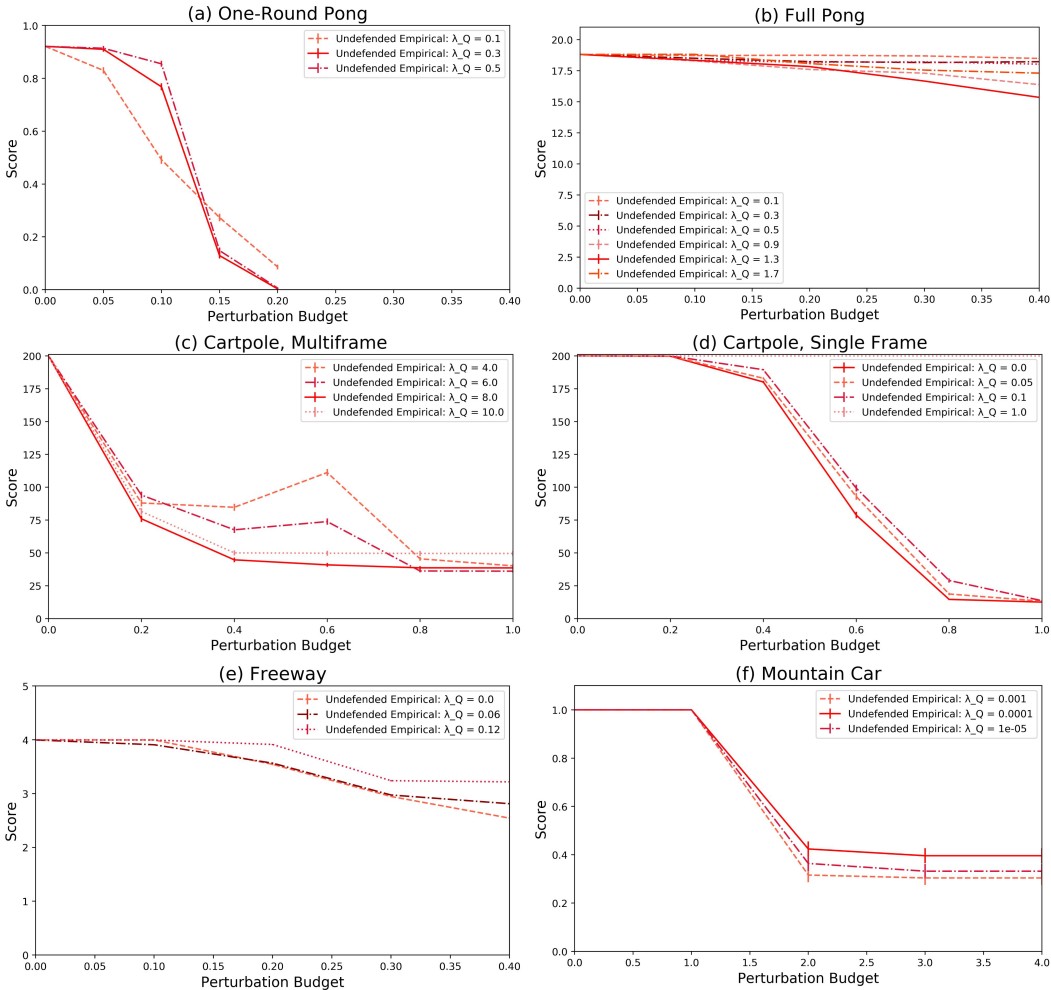

Figure 11: Empirical robustness of undefended agents on for all tested values of $\lambda_Q$ (or $\lambda$). The results in the main text are the pointwise minima over $\lambda$ of these curves.

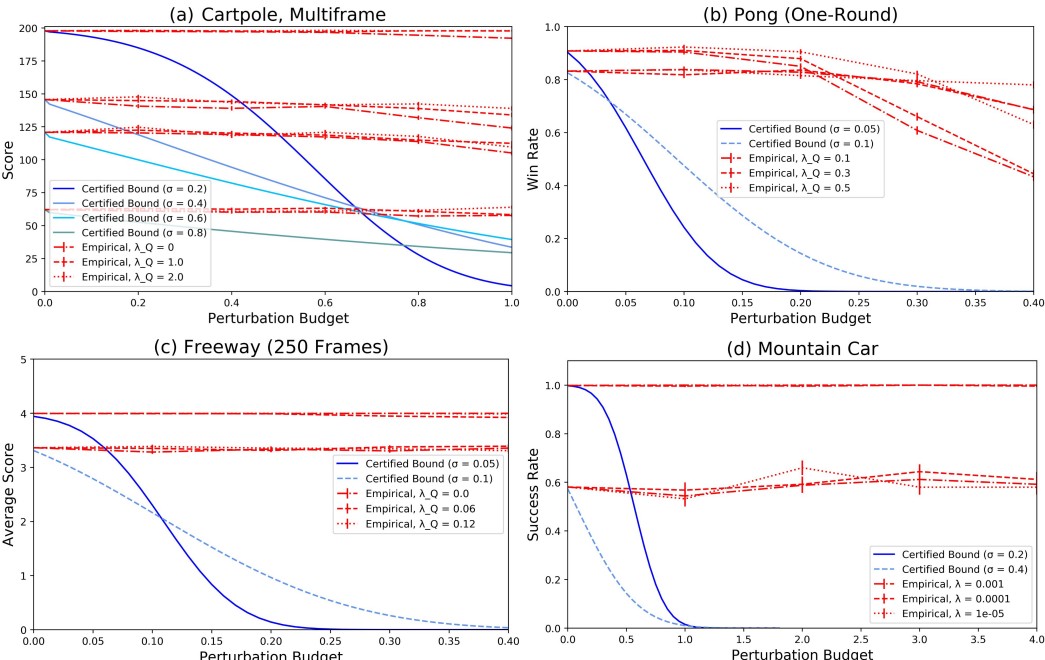

Figure 12: Empirical robustness of smoothed agents on for all tested values of $\sigma$ and $\lambda_Q$ (or $\lambda$). We also plot the associated certificate curves.

## K EMPIRICAL ATTACK DETAILS

Our empirical attack on (undefended) RL observations for DQN is described in Algorithm 1. To summarize, the core of the attack is a standard targeted $L_2$ PGD attack on the Q-value function. However, because we wish to "save" our total perturbation budget $B$ for use in later steps, some modifications are made. First, we only target actions $a$ for which the *clean-observation* Q-value is sufficiently below (by a gap given by the parameter $\lambda_Q$) the Q-value of the 'best' action, which would be taken in the absence of adversarial attack. Among these possible Targets, we ultimately choose whichever action will maximally decrease the Q-value, and which the agent can be successfully be induced to choose within the adversarial budget $B$. If no such action exists, then the original observation will be returned, and the entire budget will be saved. In order to preserve budget, the PGD optimization is stopped as soon as the "decision boundary" is crossed.

We use a constant step size $\eta$. In order to deal with the variable budget $B$, we optimize of a number of iterations which is a constant multiple $\nu$ of $\frac{B}{\eta}$.

For most environments, there is some context used by the Q-value function (i.e, the previous frames) which is carried over from previous steps, but is not directly being attacked in this round. We need both the clean version of the context, $C$, in order to evaluate the "ground-truth" values of the Q-value function under various actions; as well as the "dirty" version of the context, $C'$, based on the adversarial observations which have already been fed to the agent, in order to run the attack optimization.

Our attack for DDPG is described in Algorithm 2. Here, we use the policy $\pi$ to determine what action $a$ the agent will take when it observes a corrupted observation $o'$ (with corrupted context $C'$), and use the Q-value function supplied by the DDPG algorithm to determine the "value" of that action on the ground-truth observation $o$. Because our goal is to minimize this value, this amounts to minimizing $Q(C; o, \pi(C'; o'))$. In order to ensure that a large amount of $L_2$ "budget" is only used when the $Q$ value can be substantially minimized, we include a regularization term $\lambda\|o - o'\|_2^2$.

Attacks on smoothed agents are described in Appendix M.

| | 1-Round Pong | Full Pong | Multiframe Cartpole | Single-frame Cartpole | Freeway |
|---|---|---|---|---|---|
| Training discount factor $\gamma$ | 0.99 | 0.99 | 0.99 | 0.99 | 0.99 |
| Total timesteps | 10000000 | 10000000 | 500000 | 500000 | 10000000 |
| Validation interval (steps) | 100000 | 100000 | 2000 | 2000 | 100000 |
| Validation episodes | 100 | 10 | 10 | 10 | 100 |
| Learning Rate | 0.0001 | 0.0001 | 0.0001 | 0.00005 | 0.0001 |
| DQN Buffer Size | 10000 | 10000 | 100000 | 100000 | 10000 |
| DQN steps collected before learning | 100000 | 100000 | 1000 | 1000 | 100000 |
| Fraction of steps for exploration (linearly decreasing exp. rate) | 0.1 | 0.1 | 0.16 | 0.16 | 0.1 |
| Initial exploration rate | 1 | 1 | 1 | 1 | 1 |
| Final exploration rate | 0.01 | 0.01 | 0 | 0 | 0.01 |
| DQN target update interval (steps) | 1000 | 1000 | 10 | 10 | 1000 |
| Batch size | 32 | 32 | 1024 | 1024 | 32 |
| Training interval (steps) | 4 | 4 | 256 | 256 | 4 |
| Gradient descent steps | 1 | 1 | 128 | 128 | 1 |
| Frames Used | 4 | 4 | 5 | 1 | 4 |
| Training Repeats | 1 | 1 | 1 | 1 | 5 |
| Architecture | CNN* | CNN* | MLP $20\times 256\times 256\times 2$ | MLP $4\times 256\times 256\times 2$ | CNN* |

Table 1: Training Hyperparameters for DQN models. *CNN refers to the 3-layer convolutional network defined by the CNNPolicy class in stable-baselines3 (Raffin et al., 2019), based on the CNN architecture used for Atari games by Mnih et al. (2015). Note that hyperparameters for Atari games are based on hyperparameters from the stable-baselines3 Zoo package (Raffin, 2020), for a slightly different (more deterministic) variant of the Pong environment.

Note that on image data (i.e., Pong), we do not consider integrality constraints on the observations; however, we do incorporate box constraints on the pixel values. We also incorporate box constraints on the kinematic quantities when attacking Mountain Car, but not when attacking Cartpole: the distinction is that the constraints in Mountain Car represent artificial constraints on the kinematics [i.e., the velocity of the car is arbitrarily clipped], while the constraints in Cartpole arise naturally from the problem setup.

## L  ENVIRONMENT DETAILS AND HYPERPARAMETERS

For Atari games, we use the "NoFrameskip-v0" variations of these environments with the standard "AtariPreprocessing" wrapper from the OpenAI Gym (Brockman et al., 2016) package: this provides This environment also injects non-determinism into the originally-deterministic Atari games, by adding randomized "stickiness" to the agent's choice of actions – without this, the state-observation robustness problem could be trivially solved by memorizing a winning sequence of actions, and ignoring all observations at test-time.

Due to instability in training, for the freeway environment, we trained each model five times, and selected the base model based on the performance of validation runs. See training hyperparameters, Tables 1 and 2. For attack hyperparameters, see Table 3 and 4.

|  | Mountain Car |
|---|---|
| Training discount factor $\gamma$ | 0.99 |
| Total timesteps | 300000 |
| Validation interval (steps) | 2000 |
| Validation episodes | 10 |
| Learning Rate | 0.0001 |
| DDPG Buffer Size | 1000000 |
| DDPG steps collected before learning | 100 |
| Batch size | 100 |
| Update coefficient $\tau$ | 0.005 |
| Train frequency | 1 per episode |
| Gradient steps | = episode length |
| Training action noise | Ornstein Uhlenbeck ($\sigma = 0.5$) |
| Architecture | MLP $2 \times 400 \times 300 \times 1$ |

Table 2: Training Hyperparameters for DDPG models. Hyperparameters are based on hyperparameters from the stable-baselines3 Zoo package (Raffin, 2020), for the unmodified Mountain Car environment.

|  | 1-Round Pong | Full Pong | Multiframe Cartpole | Single-frame Cartpole | Freeway |
|---|---|---|---|---|---|
| Attack step size $\eta$ | 0.01 | 0.01 | 0.01 | 0.01 | 0.01 |
| Attack step multiplier $\nu$ | 2 | 2 | 2 | 2 | 2 |
| Q-value thresholds $\lambda_Q$ searched | .1, .3, .5 | .1, .3, .5, .9, 1.3, 1.7 | 4,6,8,10 | 0, .05, .1, 1 | 0, .06, .12 |

Table 3: Attack Hyperparameters for DQN models.

|  | Mountain Car |
|---|---|
| Attack step size $\eta$ | 0.01 |
| Attack steps $\tau$ | 100 |
| Regularization values $\lambda$ searched | .001, .0001, .00001 |

Table 4: Attack Hyperparameters for DDPG models.

---

**Algorithm 1:** Empirical Attack on DQN Agents

---

**Input:** Q-value function $Q$, clean prior observation context $C$, adversarial prior observation context $C'$, observation $o$, budget $B$, Q-value threshold $\lambda_Q$, step size $\eta$, step multiplier $\nu$

**Output:** Attacked observation $o_{\text{worst}}$, remaining budget $B'$.

$Q_{\text{clean}} := \max_{a \in A} Q(C; o, a)$

Targets $:= \{a \in A | Q(C; o, a) \leq Q_{\text{clean}} - \lambda_Q\}$

$Q_{\text{worst}} := Q_{\text{clean}}$

$o_{\text{worst}} := o$

**for** $a \in$ *Targets* **do**

    $o' := o$

    inner:

    **for** $i$ *in 1, ...,* $\lfloor \frac{\nu B}{\eta} \rfloor$ **do**

        **if** $\arg \max_{a'} Q(C'; o', a') = a$ **then**

            **if** $Q(C; o, a) < Q_{\text{worst}}$ **then**

                $o_{\text{worst}} := o'$

                $Q_{\text{worst}} := Q(C; o, a)$

            **end**

            **break** inner

        **end**

        $D := \nabla_{o'} \log([\text{SoftMax}(Q(C'; o', \cdot)]_a)$

        $o' := o' + \frac{\eta D}{\|D\|_2}$

        **if** $\|o' - o\|_2 > B$ **then**

            $o' := o + \frac{B}{\|o'-o\|_2}(o' - o)$

        **end**

    **end**

**end**

**return** $o_{\text{worst}}, \sqrt{B^2 - \|o_{\text{worst}} - o\|_2^2}$

---

**Algorithm 2:** Empirical Attack on DDPG Agents

---

**Input:** Q-value function $Q$, policy $\pi$, clean prior observation context $C$, adversarial prior observation context $C'$, observation $o$, budget $B$, weight parameter $\lambda$, step size $\eta$, step count $\tau$

**Output:** Attacked observation $o_{\text{worst}}$, remaining budget $B'$.

$o' := o$

**for** $i$ *in 1, ...,* $\tau$ **do**

    $D := \nabla_{o'}[Q(C; o, \pi(C'; o')) + \lambda\|o' - o\|_2^2]$

    **if** $\frac{\|D\|_2}{\|o'\|_2} \leq 0.001$ **then**

        **break**

    **end**

    $o' := o' + \frac{\eta D}{\|D\|_2}$

    **if** $\|o' - o\|_2 > B$ **then**

        $o' := o + \frac{B}{\|o'-o\|_2}(o' - o)$

    **end**

**end**

**return** $o', \sqrt{B^2 - \|o' - o\|_2^2}$

---

## M    ATTACKS ON SMOOTHED AGENTS

In order to attack smoothed agents, we adapted Algorithms 1 and 2 using techniques suggested by Salman et al. (2019) for attacking smoothed classifiers. In particular, whenever the Q-value function is evaluated or differentiated, we instead evaluate/differentiate the mean output under $m = 128$ smoothing perturbations. Following Salman et al. (2019), we use the same noise perturbation vectors at each step during the attack. In the multi-frame case, for the "dirty" context $C'$, we include the actually-realized smoothing perturbations used by the agents for previous steps. However, when determining the "clean" Q-values $Q(C; o, a)$, for the "clean" context $C$, we use the unperturbed previous state observations: we then take the average over $m$ smoothing perturbations of both $C$ and $o$ to determine the clean Q-values. This gives an unbiased estimate for the Q-values of an undisturbed smoothed agent in this state.

When attacking DDPG, in evaluating $Q(C; o, \pi(C'; o'))$, we average over smoothing perturbations for both $o$ and $o'$, in addition to $C$: this is because both $\pi$ and $Q$ are trained on noisy samples. Note that we use independently-sampled noise perturbations on $o'$ and $o$.

Our attack does not appear to be successful, compared with the lower bound given by our certificate (Figures 12). One contributing factor may be that attacking a smoothed *agent* is more difficult that attacking a smoothed *classifier*, for the following reason: a smoothed classifier evaluates the expected output at test time, while a smoothed agent does not. Thus, while the *average* Q-value for the targeted action might be greater than the *average* Q-value for the clean action, the actual realization will depend on the specific realization of the random smoothing vector that the agent actually uses.

## N    RUNTIMES AND COMPUTATIONAL ENVIRONMENT

Each experiment is run on an NVIDIA 2080 Ti GPU. Typical training times are shown in Table 5. Typical clean evaluation times are shown in Table 6. Typical attack times are shown in Table 7.

| Experiment | Time (hours) |
|---|---|
| Pong (1-round) | 11.1 |
| Pong (Full) | 12.0 |
| Cartpole (Multi-frame) | 0.27 |
| Cartpole (Single-frame) | 0.32 |
| Freeway | 14.2 |
| Mountain Car | 0.63 |

Table 5: Training times

| Experiment | Time (seconds): smallest noise $\sigma$ | Time (seconds): largest noise $\sigma$ |
|---|---|---|
| Pong (1-round) | 0.46 | 0.38 |
| Pong (Full) | 3.82 | 4.65 |
| Cartpole (Multi-frame) | 0.20 | 0.13 |
| Cartpole (Single-frame) | 0.18 | 0.12 |
| Freeway | 1.36 | 1.35 |
| Mountain Car | 0.67 | 0.91 |

Table 6: Evaluation times. Note that the times reported here are *per episode*: in order to statistically bound the mean rewards, we performed 10,000 such episode evaluations for each environment.

## O    CDF SMOOTHING DETAILS

Due to the very general form of our certification result ($h(\cdot)$, as a 0/1 function, can represent any outcome, and we can bound the lower-bound the probability of this outcome), there are a variety of

| Experiment | Time (seconds): smallest budget $B$ | Time (seconds): largest budget $B$ |
|---|---|---|
| Pong (1-round) | 1.01 | 0.68 |
| Pong (Full) | 8.84 | 10.2 |
| Cartpole (Multi-frame) | 0.35 | 0.32 |
| Cartpole (Single-frame) | 0.79 | 0.56 |
| Freeway | 2.67 | 2.80 |
| Mountain Car | 44.0 | 19.6 |

Table 7: Attack times. Note that the times reported here are *per episode*: in the paper, we report the mean of 1000 such episodes.

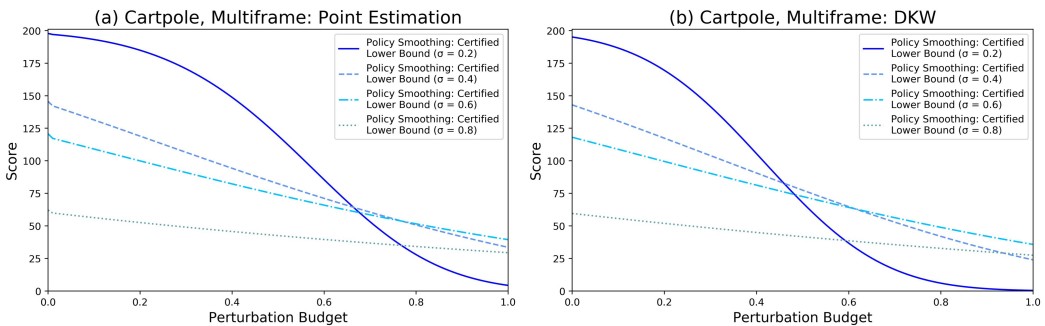

Figure 13: Comparison of certified bounds on the total reward in Cartpole, using (a) point estimation, and (b) the DKW inequality to generate empirical bounds.

ways we can use the basic result to compute a certificate for an entire episode entire game. In the main text, we introduce CDF smoothing Kumar et al. (2020) as one such option. In CDF smoothing for any threshold value $x$, we can define $h_x(\cdot)$ as an indicator function for the event that the total episode reward is greater than $x$. Then, by the definition of the CDF function, the expectation of $h_x(\cdot)$ is equal to $1 - F(x)$, where $F(\cdot)$ is the CDF function of the reward. Then our lower-bound on the expectation of $h_x(\cdot)$ under adversarial attack is in fact an upper-bound on $F(x)$: combining this with Equation 2 in the main text,

$$E[\mathcal{X}] = \int_0^\infty (1 - F(x))dx - \int_{-\infty}^0 F(x)dx,$$

provides a lower bound on the total expectation of the reward under adversarial perturbation.

However, in order to perform this integral from empirical samples, we must bound $F(x)$ at all points: this requires first upper-bounding the *non-adversarial* CDF function at all $x$, before applying our certificate result. Following Kumar et al. (2020), we accomplish this using the Dvoretzky–Kiefer–Wolfowitz inequality (for the Full Pong environment.)

In the case of the Cartpole environment, we explore a different strategy: note that the reward at each timestep is itself a 0/1 function, so we can define $h_t(\cdot)$ as simply the reward at timestep $t$. We can then apply our certificate result at each timestep independently, and take a sum. Note that this requires estimating the average reward at each step independently: we use the Clopper-Pearson method (following Cohen et al. (2019)), and in order to certify in total to the desired 95% confidence bound, we certify each estimate to (100 - 5/T)% confidence, where $T$ is the total number of timesteps per episode (= 200).

However, note that, in the particular case of the cartpole environment, $h_t(\cdot) = 1$ if and only if we have "survived" to time-step $t$: in other words, $h_t(\cdot)$ is simply an indicator function for the total reward being $\geq t$. Therefore in this case, this independent estimation method is equivalent to CDF smoothing, just using Clopper-Pearson point-estimates of the CDF function rather than the Dvoretzky–Kiefer–Wolfowitz inequality. In practice, we find that this produced slightly better certificates for this task. (Figure 13)

## P    ENVIRONMENT LICENSES

OpenAI Gym Brockman et al. (2016) is Copyright 2016 by OpenAI and provided under the MIT License. The stable-baselines3 packageRaffin et al. (2019) is Copyright 2019 by Antonin Raffin and also provided under the MIT License.

