# OpenReview forum: "Policy Smoothing for Provably Robust Reinforcement Learning"
_ICLR.cc/2022/Conference — ICLR 2022 Poster_

### Official Review · Reviewer_LDzS · 2021-10-25

**Correctness:** 4
**Technical Novelty And Significance:** 3
**Empirical Novelty And Significance:** 3
**Recommendation:** 6
**Confidence:** 2

**Main Review:**

1. In the adversarial objective defined above equation (1), it appears as if the perturbation sequence $\epsilon_t$ is fixed regardless of what actual state $s_t$ is observed, whereas from later sections it's obvious that the paper actually allows $\epsilon_t$ to depend on $s_t$.

2. What is \Phi in Theorem 1?

3. In the experiments, what exactly is the undefended agent? It is stated that the defense applies policy smoothing during both training and testing phases. So for the construction of undefended agents, are they constructed just via standard training, i.e. with no smoothing during training either?

The result in Figure 5 seems unintuitive because I would anticipate the undefended agent to perform somewhat better than the smoothed policy when there is no adversarial perturbation, as intuitively techniques such as smoothing must deteriorate the performance in a clean environment.

I'm not very familiar with the Neyman-Pearson Lemma, but it seems unintuitive to me that in Lemma 3, one can provide a robustness guarantee against an adaptive adversary from the construction of a much weaker adversary. Some remarks and background regarding the Neyman-Pearson Lemma would be helpful.


**Summary Of The Paper:**

This paper extends the smoothing technique to the RL setting as a defense to test-time adversarial attacks. An adaptive version of the Neyman-Pearson Lemma is proposed. Empirical evaluations showcase the effectiveness of policy smoothing.

**Summary Of The Review:**

Experiment results seem convincing, except for some minor issues. I'm not familiar with the prior literature to judge the significance of the new Neyman-Pearson Lemma. The exposition can certainly be improved to make the paper more self-contained and readable.

---

> ### Author Response · Authors · 2021-11-18
> **Authors' Response**
>
> Thank you for your feedback and constructive suggestions. We address your comments below:
>
> - **"In the adversarial objective defined above equation (1), it appears as if ..."** -- Thank you for pointing this out. Indeed, the adversarial perturbations are allowed to depend on the current state.
>     They may also depend on previous states and actions generated in the game.
>     We have included this clarification in the 'Problem setup' section.
>
> - **"What is $\Phi$ in Theorem 1?"** -- $\Phi$ is the cumulative distribution function of the standard normal distribution. We have included this definition in the theorem.
>
> - **"In the experiments,  what exactly is the undefended agent? [...] are they constructed just via standard training, i.e. with no smoothing during training either?"** -- Yes, the undefended agent is simply an agent trained in the standard way, with no smoothing during training or testing.
>
> - **"The result in Figure 5 seems unintuitive ..."** -- As shown in Figure 4 (and now additionally in Figure 9 in the appendix, which we have added) there is indeed a deterioration in "clean" performance as the magnitude of smoothing noise $\sigma$ increases. However, for low levels of smoothing noise (such as those shown in Figure 5), there does not seem to be any effect: at these low levels of noise, the agents seem to be able to gather sufficient information from the noisy observations to adequately perform the tasks.
>
> - **... it seems unintuitive to me that in Lemma 3, one can provide a robustness guarantee against an adaptive adversary from the construction of a much weaker adversary ..."** -- The structured adversary $\epsilon^{st}$ that exhausts the entire budget in the first time-step is the worst-case adversary for the worst-case environment-policy pair $(M', \pi')$ as described in Section 4.2 on the tightness of the certificate.
>     This does not imply that the structured adversary is the strongest adversary for a general environment-policy pair.
>     In the tightness analysis, we show that there exists a worst-case environment-policy pair $(M', \pi')$, over the set $H_p$ of *all* environment-policy pairs that achieve the reward threshold with probability $p$, for which the structured adversary $\epsilon^{st}$ can reduce this probability to $\Phi ( \Phi^{-1}(p) - B/\sigma)$.
>     In lemma 3, we essentially show that the performance of $\pi'$ in $M'$ under $\epsilon^{st}$ is a lower bound on the performance (under an adversary) of a general policy in a general environment.
>     We construct the adversary $\epsilon^{st}$ mainly for theoretical analysis.
>
> - **"Some remarks and background regarding the Neyman-Pearson Lemma would be helpful."** -- The Neyman-Pearson lemma is a theory in statistical hypothesis testing that determines the significance of one hypothesis over another based on a specified property of an observed sample.
>     In the context of randomized smoothing in the static setting, it produces the worst-case decision boundary of a classifier based on the estimated probability of the top class under the smoothing distribution.
>     It says that the worst-case decision boundary is a region where the ratio of the probability density functions of the smoothing distributions at the clean input and the perturbed input is a constant.
>     When the two distributions are isometric Gaussians, as is the case for static problems like image classification under Gaussian smoothing, this boundary takes the form of a hyperplane leading to the robustness bounds in Theorem 1 of [1] (see appendix A of [1]).
>     However, in the dynamic setting of RL, the smoothing distribution after adding the adversarial perturbation may not be isometric even if the smoothing noise at each time-step was sampled from an isometric Gaussian distribution (see figure 1, section `Technical contributions' on page 2 and appendix A of our manuscript).
>     This is why we formulate and prove an adaptive version of the Neyman-Pearson lemma to obtain provable robustness through randomized smoothing.
> We have added this discussion in the updated manuscript.
>
>     [1] Certified Adversarial Robustness via Randomized Smoothing, Cohen et. al., ICML 2019.

---

### Official Review · Reviewer_6S68 · 2021-11-01

**Correctness:** 4
**Technical Novelty And Significance:** 3
**Empirical Novelty And Significance:** 3
**Recommendation:** 8
**Confidence:** 4

**Main Review:**

I am glad to see RL-verification approach that considers traces, rather than just applying static verification techniques to individual time-steps.
While the resulting algorithm is a straight-forward adaption of RS, I think the conceptual and mathematical arguments on how to apply it in this setting are the key strength of the paper.
Further, the main paper and the technical arguments in appendices B - D are well written and mostly easy to follow. The key mathematical results seem to be correct.

That said, the results indicate that the presented approach does not provide a usable defense for sequential-decision-making yet, as none achieved score-vs-budget trade-offs seems really desirable. While not ideal, I think the paper can still benefit the community by shifting focus on trace-based certificates in sequential decision making and is further valuable for its theoretical contributions.

To conclude I have a few questions:
- In the tightness proof (section 4.2) you suddenly assume another policy $\pi'$. Can you elaborate as to why the argument isn't over the initial $\pi$ and why this change is possible?
- Do you think it would be possible to utilize multiple temporally overlapping certificates to improve upon the budget-score trade-off?
- [1] seems to provide similar guarantees to the proposed approach. Can you briefly outline the difference in approach?

[1] CROP: Certifying Robust Policies for Reinforcement Learning through Functional Smoothing; Fan Wu et al.; arXiv 2021


**Summary Of The Paper:**

The paper discusses the extension of randomized smoothing (RS) to sequential interactive settings, on the example of reinforcement learning (RL).
The authors show, that in order to certify a sequence of steps in this setting it the theory from RS can not be directly utilized as it does not account for the adaptiveness of the adversary/environment.
To enable the application of RS, the authors propose an adaptive version of the Neyman-Pearson Lemma, which can be used to instantiate RS to this setting.
Mathematically, this works by first showing that the impact of a potentially stochastic adversary can be bounded by a deterministic adversary, which in turn can be bounded by a deterministic adversary with special structure.

Equipped with this theory, the authors introduce policy smoothing, an algorithm that lower-bounds on the reward (or other functions of the trace) that an agent will obtain (with high probability) in the presence of an adversary.
The authors showcase the effectiveness of the approach on the Cartpole, Mountain Car, Pong and Freeway for different deep RL agents.

**Summary Of The Review:**

The paper presents a novel way to show robustness to perturbations in sequential decision making in the presence of an adversary.
While empirical results are not strong, the paper appears to be formally correct and would benefit the community.

---

> ### Author Response · Authors · 2021-11-18
> **Authors' Response**
>
> Thank you for your succinct and accurate summarization of our work. We really appreciate you highlighting the importance of trace-based certificates. We address your main comments in the following:
>
> - **"... the results indicate that the presented approach does not provide a usable defense for sequential-decision-making yet ..."** -- Our certificates depend on the combined $\ell_2$-norm of all the perturbations added in an episode.
>     This is perhaps not ideal for long games that run for a large number of time-steps.
>     As our tightness result shows, this is the best one can do using randomized smoothing without making any assumptions on the policy or the environment.
>     However, if we can make some assumptions about the environment and the policy, say, the state of the environment in one step is not affected by the actions of the previous steps, better score-vs-budget trade-offs might be possible.
>
> - **"In the tightness proof (section 4.2) you suddenly assume another policy $\pi'$. Can you elaborate as to why the argument isn't over the initial $\pi$ and why this change is possible?"** -- In the tightness analysis, we show that there exists a worst-case environment-policy pair $(M', \pi')$, over the set $H_p$ of *all* environment-policy pairs that achieve the reward threshold with probability $p$, for which the structured adversary $\epsilon^{st}$ can reduce this probability to $\Phi ( \Phi^{-1}(p) - B/\sigma)$.
>     We are not just changing the policy but also the environment and showing that the performance of $\pi'$ in $M'$ under $\epsilon^{st}$ is a lower bound on the probability achieved by a general environment-policy pair under a general adversary.
>     This does not mean that the structured adversary is the strongest adversary for a general environment-policy pair.
>     Instead, our theoretical results show that the probability $\Phi ( \Phi^{-1}(p) - B/\sigma)$ achieved by $\epsilon^{st}$ for $(M', \pi')$ is a lower-bound on the probability achieved by a general adversary $\epsilon$ for a general environment-policy pair $(M, \pi)$.
>     The reason why we transform the environment-policy pair to this special case $(M', \pi')$ is to eliminate the dynamic and adaptive nature of adversarial RL and cast it into the static setting.
>
> - **"Do you think it would be possible to utilize multiple temporally overlapping certificates to improve upon the budget-score trade-off?"** -- Given our tightness results, it does not seem possible to achieve a better budget-score trade-off without making any assumptions about the policy function or the dynamics of the environment.
>     However, if we are allowed to make assumptions such as the states are independent of the actions picked, it might be possible to get better trade-offs.
>
> - **"[1] seems to provide similar guarantees to the proposed approach. Can you briefly outline the difference in approach?"** -- While the robustness bound in theorem 3 in Wu et. al. 2021 is similar to ours, the proof in [1] does not account for the adaptive nature of the RL adversary.
>     Instead, it applies the proof from Chiang et. al. [2] which is for the static setting.
>     The proof, in the static setting, relies on the fact that the smoothing distribution remains an isometric Gaussian even after adding the adversarial perturbation.
>     This leads to a worst-case classifier which is a half-space. This property of the worst-case classifier is needed to derive the bound of the form: $\Phi ( \Phi^{-1}(p) - \epsilon \sqrt{H}/\sigma)$.
>     However, as we illustrate in Figure 1, the smoothing distribution after adding the adversarial perturbation in the RL setting need *not* be an isometric Gaussian distribution and the worst-case decision boundary will no longer be a hyper-plane.
>     We discuss this challenge in detail under 'Technical contributions' on page 2 and in Appendix A.
>     Applying the proof directly from the static setting does *not* work for the dynamic setting of RL. This is why we formulate and prove an adaptive version of the Neyman-Pearson lemma to adapt randomized smoothing in the dynamic setting of RL which is the main contribution of our work.
>
>     [2] Detection as regression: Certified object detection with median smoothing, Chiang et. al., NeurIPS 2020.

---

> > ### Comment · Reviewer_6S68 · 2021-11-22
> > **Response**
> >
> > Dear Authors,
> >
> > Thank you for clarifying these points.
> > As this addresses all open questions I had, I have raised my score.
> > While not ultimately necessary, I would appreciate if section 4.2 outlined the high-level idea, similar to what you wrote here, before showing the actual constructions.
> >
> > Best Regards,
> > Reviewer 6S68

---

> > > ### Author Response · Authors · 2021-11-23
> > > **Thank you for the score increase!**
> > >
> > > We are glad we were able to address your concerns. We have included the above discussion about the tightness result in section 4.2 of the updated draft right before the construction of the worst-case environment-policy pair.
> > >
> > > Thank you again for your valuable feedback! Your suggestions have greatly improved our paper.

---

### Official Review · Reviewer_4eym · 2021-11-02

**Correctness:** 3
**Technical Novelty And Significance:** 3
**Empirical Novelty And Significance:** 3
**Recommendation:** 6
**Confidence:** 5

**Main Review:**

Pros:
Regularization via augmentation with noisy samples has been a major focus of research in defending against inference-time adversarial perturbations. This paper advances the state of the art by presenting tight lower-bounds on the return of policies that are trained on randomly perturbed observations, when subjected to norm-limited adversarial perturbations of observations. This work also overcomes the limitations of prior work by Zhang et al. via extending the certification to the entire trajectory, and not just step-wise experiences of the agent.
The theoretical treatment of the problem is sound, and clearly stated.
Cons:
In multiple sections of the manuscript, authors claim that the proposal of policy smoothing is a novel contribution of their work. This does not seem to be the case, as numerous works in the literature have already proposed and investigated the regularization of RL policies via training on randomly and adversarially perturbed observations.
In the entirety of the presented analysis, there seems to be an implicit assumption that the adversary is aware of the observations of the target agent. This assumption is expressed more clearly in Algorithms 1 and 2. Do the authors assume that in the absence of the adversary, the target agent’s observation would be the same as the state? If so, this needs to be clearly stated in both the problem setup and the threat model.
The empirical results seem lacking. If policy smoothing is a focus of this study, the results should also report the effect of policy smoothing at both training and test-time under normal (i.e., non-adversarial) settings. This requires more information to be presented than just the single dot on the performance curves under adversarial attacks with 0 budget.
In the full Pong game, the certified lower-bound seems to be very loose compared to the presented results. While the interpretation presented in the appendix seems valid, it may be best to rethink the attack methodology to show that the lower-bound is meaningful.
Minor comments:

Page 2 - below equation 1: “...end after t time steps” - perhaps would be better to use upper-case T to avoid confusion in symbols.

Page 8: “an attack tailored to the threat model defined in 1”


**Summary Of The Paper:**

The paper presents theoretical lower bounds (i.e., certified robustness guarantees) on the performance of RL policies with randomized smoothing against norm-limited adversaries. The proposal of the paper is an extension of the Neyman-Pearson Lemma to the adaptive settings. Authors present formal proofs for their proposal, and evaluate their claims in 4 RL benchmarks.

**Summary Of The Review:**

I find this paper to be a valuable contribution to the field, and believe that the theoretical treatment and methodology of this paper can inform future investigations in not only test-time adversarial robustness, but also the general analyses of sequential perturbations in RL.

---

> ### Author Response · Authors · 2021-11-18
> **Authors' Response**
>
> Thank you for your encouraging comments and recognition of our work as "a valuable contribution to the field."
> We really appreciate your detailed feedback and constructive suggestions. Below, we address your main comments:
>
> - **Novelty of our work:** While procedures that augment the dynamics of the RL system with random or adversarial noise to train robust policies have been studied, our procedure produces guarantees on the expected total reward obtained by the smoothed policy.
>     To the best of our knowledge, previous approaches do not produce certified robustness guarantees on the expected total reward as a function of the adversarial noise budget over the entire trajectory.
>     The main contribution of our work is to derive robustness certificates for randomized smoothing in the dynamic setting of reinforcement learning.
>
> - **"Do the authors assume that in the absence of the adversary, the target agent’s observation would be the same as the state?"** -- No, we model the RL framework as a Partially Observable Markov Decision Process where the target agent only sees the *observations* of the states regardless of whether an adversary is present or not.
>     We have clarified this under 'Problem setup' on page 2 of the updated manuscript.
>     The case where all the $\epsilon_i$s are equal to zero denotes the absence of the adversary. The problem formulation, in this case, remains the same as the general case with non-zero $\epsilon_i$s.
>
> - **"... there seems to be an implicit assumption that the adversary is aware of the observations of the target agent."** -- We allow the adversary to have access to the observations of the agent to certifiably defend the agent policy against the strongest adversary that we can.
>     Our goal is to produce certified robustness for RL policies in the most general setting that we can.
>     Our procedure and certificates do not make any assumptions about the policy or the environment, and permit the adversary to have as much information as possible.
>
> - **"If policy smoothing is a focus of this study, the results should also report the effect of policy smoothing at both training and test-time under normal (i.e., non-adversarial) settings."** -- This is now provided in Appendix G, where we report average 'clean' (no adversarial perturbation) test-time scores as a function of smoothing noise for all experiments in the main text, as well as validation score curves from training.

---

### Official Review · Reviewer_RqNT · 2021-11-02

**Correctness:** 4
**Technical Novelty And Significance:** 3
**Empirical Novelty And Significance:** 3
**Recommendation:** 6
**Confidence:** 3

**Main Review:**

> In general, I think this paper is clearly written and easy to follow. The proposed method is intuitive and the experimental results are convincing. Studies on RL robustness are still limited and many questions remain open.

> A major competitor of this paper is Zhang et al, NeurIPS 2020, which reduced the significance of contribution. But I agree that the authors adequately discussed the connection and difference between their work and the related works. Also, though the philosophy is borrowed from Cohen et al, randomized policy smoothing is still a completely new approach for defending RL models.

> I would also point out that the formulation of $\bar{\pi}$ in Section 4.3 is very similar to a POMDP and I guess that the authors did not realize this. I think it would be better if the authors can provide some discussions on this matter. Similar connections are also mentioned in Zhang, et al. "Robust reinforcement learning on state observations with learned optimal adversary." ICLR 2021, which I think the authors may also need to include in the reference.

**Summary Of The Paper:**

This paper proposed a method for RL agents' robustness certification for using randomized smoothing. An adaptive Neyman-Pearson Lemma is developed and a robustness guarantee in terms of the cumulative reward is obtained. This paper also gives a worst-case setting to prove the tightness of the aforementioned certification and also evaluate their certificates in two environments under adversarial attacks.

**Summary Of The Review:**

This paper studies an important and novel problem with limited works in the literature: RL agents' robustness certification. The authors made strong assumptions and borrow ideas (randomized smoothing) from existing works in similar fields. The contribution is obvious, though not very significant. I am positive about this paper and recommend a weak acceptance.

---

> ### Author Response · Authors · 2021-11-17
> **Authors' Response**
>
> Thank you for your valuable feedback! We really appreciate your constructive suggestions regarding the problem formulation.
> We address your main concerns below:
>
> - **"... the formulation of $\bar{\pi}$ in Section 4.3 is very similar to a POMDP ..."** -- Indeed, the reinforcement learning framework we consider can be expressed as a Partially Observable Markov Decision Process (POMDP) which we have clarified in the updated draft.
>     Our results, therefore, apply to a generalization of an MDP where the agent may only partially observe the state $s_t$ of the environment through the observation function $o(s_t)$ (page 2, under 'Problem Setup').
>     Moreover, our procedure does not make any assumptions on the policy or the environment and only requires black-box access to those.
>     In Section 3, we model the entire adversarial RL process under Policy Smoothing as a sequence of interactions between a system **A**, which captures the RL environment and the agent, and a system **B**, which captures the addition of the adversarial perturbation and the smoothing noise.
>     Our theoretical results do not require these systems to be Markovian and can thus have potential applications in real-time decision-making processes that do not necessarily satisfy the Markov property. We have included this discussion in the paper.
>
> - **"Similar connections are also mentioned in Zhang, et al. ..."** -- Thank you for bringing Zhang et al.'s ICLR 2021 work to our notice.
>     We have included this work in the references of our paper.
>     It proposes a very novel adversarial training procedure for deep reinforcement learning policies by training them in an online alternating fashion with a learned adversary.
>     They show that finding an optimal agent policy under a fixed adversary is a POMDP problem.
>     They achieve better empirical robustness than existing methods of adversarial training.
>     Our procedure, on the other hand, generates provable robustness guarantees on the performance of the smoothed policy.
>     It can be applied to any policy regardless of how it has been trained.
>     For instance, robust policies from Zhang et. al.'s work can be smoothed using our procedure to obtain certified guarantees on their performance in the presence of an adversary.

---

### Official Review · Reviewer_xL3X · 2021-11-08

**Correctness:** 3
**Technical Novelty And Significance:** 2
**Empirical Novelty And Significance:** 2
**Recommendation:** 5
**Confidence:** 2

**Main Review:**

Strengths:
This paper tries to tackle an important problem: how to get certified robustness in reinforcement learning (RL) problems. The method that the authors propose brings the gap between the randomized smoothing technique and RL.

Weakness:
I think this paper needs to be better polished in order to be published. I found several places that are quite hard to follow.
I also find that the idea is quite incremental given that randomized smoothing has been widely studied in supervised learning and the key idea in this paper is quite similar. The authors mentioned that the Adaptive Neyman-Pearson Lemma is a non-trivial result, but I found that the discussion about this lemma is quite confusing.

Below are some detailed comments.

I found Figure 1 quite hard to understand. It would be helpful to explain this figure better.

I found that the problem formulation is a bit inconsistent throughout the paper. In Section 1, it seems to me that the adversary can only add $\epsilon_i$ to the observation. Then I found that in Section 3, it is mentioned that the adversary can also choose a smoothing noise $\delta_i$. It would be useful to clarify and make the formulation consistent. Is the distribution of $\delta_i$ chosen before the algorithm starts? Does the adversary have to add $\delta_i$ in the perturbation? For example, if the adversary believes that the $\epsilon_i$ perturbation is strong enough, maybe it can skip adding $\delta_i$?

Lemma 1: the symbol S has already been used to denote the state space.

I am a bit confused about the construction of $Y^{st}$. Does it mean that the best strategy for the adversary is to put all the perturbation budget to the first time step? Is this phenomenon observed in the experiments or is it only for theoretical purpose?

Section 4.3: can the authors provide more specific expressions for "underline F" and "overline F"?

===After response===

Thanks to the authors for the response. Now I understand that system B is not the adversary; instead it is a virtually constructed system for the purpose of the proof. This makes more sense and helped me understand the paper better. However, it should be more explicitly explained in the paper what is the actual attack system (which part the adversary is allowed to perturb) and what is the virtual system used for analysis. Currently in Figure 2 of the paper, it still seems to me that system B is just the attacker.

I increased my score but overall I still feel that there is a lot of room for improvement in terms of the clarify of the paper.

I also feel that I am not familiar enough with the existing proof technique for randomized smoothing so I may not be able to evaluate the correctness very well. Thus I decided to lower my confidence score as well.

**Summary Of The Paper:**

This paper studies certified robustness of a policy in a reinforcement learning setting. The authors propose a method, similar to randomized smoothing in supervised learning to get certified robustness for a policy. The authors provide theoretical evidence and experimental results.

**Summary Of The Review:**

Overall, I found this paper a bit hard to follow and I don't think the key messages are clearly conveyed. I recommend the authors to carefully revise this paper in order to be published.

---

> ### Author Response · Authors · 2021-11-17
> **Authors' Response**
>
> Thank you for your detailed review and constructive suggestions. Below we address your comments one by one:
>
> - **"I also find that the idea is quite incremental given that randomized smoothing has been widely studied in supervised learning ..."** -- We respectfully disagree. The main contribution of this work is to derive robustness certificates for randomized smoothing in **the dynamic setting** of reinforcement learning which is highly non-trivial and in our opinion quite fundamental. We explain this below:
>
>     The robustness guarantees in supervised learning critically rely on the fact that the adversarial perturbations added to the input do not depend on the Gaussian smoothing noise. This ensures that the Gaussian distributions at the original input and the perturbed input are both isometric.
>     Only then will the worst-case decision boundary, governed by the Neyman-Pearson lemma, will be a hyperplane perpendicular to the direction of the perturbation.
>     The lemma says that the worst-case decision boundary is a region where the ratio of the probability distribution functions for both these distributions is constant (which takes the form of a hyper-plane in the supervised setting).
>
>     However, due to the adaptive nature of RL, the Gaussian noise added at one time-step could influence the action at that step, which in turn could affect the state produced in the next step, ultimately influencing the adversarial perturbation added in the next step.
>     Thus, the smoothing distribution after adding adversarial perturbations, i.e., the distribution of the concatenated vector $(\epsilon_1 + \delta_1, \epsilon_2 + \delta_2, \epsilon_3 + \delta_3, \ldots)$, need not be an isometric Gaussian distribution and the worst-case boundary will no longer be a hyper-plane.
>     Figure 1, especially parts (a) and (d), illustrate this challenge.
>     This is why we prove an adaptive version of the Neyman-Pearson lemma in order to derive robustness certificates in RL.
>     The proof for all the lemmas used in the derivation of the robustness certificate can be found on pages 14 - 18 in the appendix.
>
>     Our work differs from previous certified defenses in RL with respect to the objective of the robustness guarantees.
>     While previous methods have certified the per-step action taken by the agent given the observations, we can certify the *overall performance* (reward obtained) of the agent policy for the entire episode.
>     In robust RL, the ultimate goal is to obtain a policy that achieves a good total reward regardless of how much the actions at individual time-steps change.
>
> - **"I found Figure 1 quite hard to understand. It would be helpful to explain this figure better."** -- Figure 1 illustrates the main challenge in proving robustness guarantees in the RL setting as compared to the supervised setting. We have explained this figure above while addressing your previous comment.
>
> - **" ... in Section 3, it is mentioned that the adversary can also choose a smoothing noise $\delta_i$."** -- The adversary only chooses the adversarial perturbation $\epsilon_i$ and not the smoothing noise $\delta_i$.
>     In section 3, we combine the steps of the adversary adding its perturbation and the agent adding the smoothing noise into one system **B**.
>     This is not mean that the adversary adds the smoothing noise as well.
>     We use $\eta_i = \epsilon_i + \delta_i$ to denote the net change in the observation at step $i$ after adding the adversarial perturbation and the smoothing noise.
>     The rest of the RL framework, including the environment and the agent, is denoted as a separate system **A**.
>     We model the entire process as interactions between these two systems, generating tokens $\tau_i$ and offsets $\eta_i$ at each round $i$, to make it easy to formulate the desired lemmas for the robustness guarantee.
>
> - **"Is the distribution of $\delta_i$ chosen before the algorithm starts?"** -- We fix the smoothing distribution from which each $\delta_i$ is sampled to be a Gaussian with variance $\sigma^2$ (section 4.3).
>     The robustness guarantees depend on this distribution being a Gaussian.
>     However, each $\delta_i$ is sampled only after the adversary picks the perturbation $\epsilon_i$ in time-step $i$. Thus, while $\epsilon_i$ may depend on noise samples from previous steps, it cannot be a function of the one sampled in step $i$.
>     This is crucial for proving the adaptive Neyman-Pearson lemma.

---

> > ### Author Response · Authors · 2021-11-17
> > **Authors' Response contd.**
> >
> > - **"Does the adversary have to add $\delta_i$ in the perturbation?"** -- No, the adversary only adds perturbation $\epsilon_i$ to the observation and the agent adds the smoothing noise $\delta_i$ to the perturbed observation.
> >     We combine the steps of the adversary adding its perturbation and the agent adding the smoothing noise into one system **B** to make it easy to formulate the lemmas we need to derive the robustness guarantees.
> >     This is not mean that the adversary adds the smoothing noise as well.
> >     We have explained this in more detail above while addressing your previous comments.
> >
> > - **"For example, if the adversary believes that the  perturbation is strong enough, maybe it can skip adding $\delta_i$?"** -- The smoothing noise $\delta_i$ is added by the agent to defend itself against the adversary.
> >     It is not added by the adversary to strengthen its attack.
> >
> > - **"... the symbol S has already been used to denote the state space."** -- Thank you for pointing this out! We have changed the $S$ in lemma 1 and the following lemmas to $\Gamma$.
> >
> > - **Construction of $Y^{st}$:** This is the random variable corresponding to the distribution of the tuple $z = (\tau_1, \eta_1, \tau_2, \eta_2, \ldots, \tau_t, \eta_t)$ after applying the smoothing noise to the structured adversary $\epsilon^{st}$. This adversary exhausts the entire budget at the first time-step.
> >     This is the worst-case adversary for the worst-case environment-policy pair $(M', \pi')$ as described in Section 4.2 on the tightness of the certificate.
> >     This does not imply that the structured adversary is the strongest adversary for a general environment-policy pair.
> >     However, in the adaptive Neyman-Pearson lemma, we essentially show that the performance of $\pi'$ in $M'$ under $\epsilon^{st}$ is a lower bound on the performance (under an adversary) of a general policy in a general environment.
> >     We construct the adversary $\epsilon^{st}$ mainly for the theoretical results we need to derive our robustness bounds.
> >
> > - **Expressions for $\underline{F}$ and $\overline{F}$:** These functions are high-confidence bounds on the true CDF $F(x)$ using the empirical CDF $F_m(x)$ obtained using the Dvoretzky–Kiefer–Wolfowitz inequality.
> >     It says that with probability $1 - \alpha$, for $\alpha \in (0, 1]$, the true CDF $F(x)$ is in the range $[\underline{F}(x), \overline{F}(x)]$, where $\underline{F}(x) = F_m(x) - \sqrt{\ln (2/\alpha) / 2m}$ and $\overline{F}(x) = F_m(x) + \sqrt{\ln (2/\alpha) / 2m}$.
> >     We have added these expressions in the updated version of the paper.

---

> ### Author Response · Authors · 2021-11-27
> **Follow-up**
>
> Dear Reviewer xL3X:
>
> It has been a while since we posted our response to your initial review. Since the final phase of the discussion period is nearing its end, we were wondering if you have had a chance to go through our rebuttal and if we have been able to address your concerns. If we could provide further explanations regarding our work, please let us know.
>
> Thank you!
>
> -Authors

---

> > ### Comment · Reviewer_xL3X · 2021-11-30
> > **Review updated**
> >
> > Apologies for the delay. I updated my review.

---

> > > ### Author Response · Authors · 2021-12-03
> > > **Thank you for the score increase!**
> > >
> > > We appreciate your valuable feedback. We will make sure to include your suggestions in the next update of the manuscript.

---

### Author Response · Authors · 2021-11-19
**Thank you to all reviewers**

Dear Reviewers:

Thank you all for your time and effort in reviewing our work. Your feedback has greatly improved our paper. In our responses, we did our best to address all of your concerns. Since the discussion period is nearing its end, we were wondering if you had any additional comments or questions for us. If there is more we could do to help you make your final decision, please let us know.

-Authors

---

### Decision · Program_Chairs · 2022-01-20

**Decision:**

Accept (Poster)

**Comment:**

The reviewers appreciated the treatment of the topic of certifiable robustness done in this work and although they had a number of concerns, I feel they were adequately addressed by the authors.